### elife.elifesciences.org

# Shugoshin biases chromosomes for biorientation through condensin recruitment to the pericentromere

**Kitty F Verzijlbergen[1], Olga O Nerusheva[1], David Kelly[1], Alastair Kerr[1], Dean Clift[1†], Flavia de Lima Alves[1], Juri Rappsilber[1,2], Adele L Marston[1*]**

[1]Wellcome Trust Centre for Cell Biology, Institute of Cell Biology, University of Edinburgh, Edinburgh, United Kingdom; [2]Department of Biotechnology, Institute of Bioanalytics, Technische Universität Berlin, Berlin, Germany

**Abstract** To protect against aneuploidy, chromosomes must attach to microtubules from opposite poles ('biorientation') prior to their segregation during mitosis. Biorientation relies on the correction of erroneous attachments by the aurora B kinase, which destabilizes kinetochore-microtubule attachments that lack tension. Incorrect attachments are also avoided because sister kinetochores are intrinsically biased towards capture by microtubules from opposite poles. Here, we show that shugoshin acts as a pericentromeric adaptor that plays dual roles in biorientation in budding yeast. Shugoshin maintains the aurora B kinase at kinetochores that lack tension, thereby engaging the error correction machinery. Shugoshin also recruits the chromosome-organizing complex, condensin, to the pericentromere. Pericentromeric condensin biases sister kinetochores towards capture by microtubules from opposite poles. Our findings uncover the molecular basis of the bias to sister kinetochore capture and expose shugoshin as a pericentromeric hub controlling chromosome biorientation.

*For correspondence: adele.marston@ed.ac.uk

†Present address: MRC Laboratory of Molecular Biology, Cambridge, United Kingdom

Competing interests: The authors declare that no competing interests exist.

## Introduction

The accurate segregation of chromosomes during mitosis relies on the capture of newly duplicated sister chromatids by microtubules emanating from opposite poles. This is possible because on each chromosome a multi-subunit kinetochore assembles on a region of DNA, known as a centromere, to mediate attachment to microtubules. The proper attachment of sister kinetochores to opposite poles, called biorientation, generates tension owing to sister chromatid cohesion (*Tanaka, 2010*). Sister kinetochores are inherently biased towards capture by microtubules from opposite poles, yet how this is achieved is not known (*Indjeian and Murray, 2007*). Kinetochore geometry is thought to position microtubule binding sites in a 'back-to-back' orientation during mitosis and this has been hypothesized to contribute to biorientation (*Hauf and Watanabe, 2004*). Where erroneous tension-less attachments do occur, they are destabilized by the aurora B kinase, providing a further opportunity for biorientation to be established (*Tanaka, 2010*). Shugoshin proteins are localized to the region surrounding the centromere, known as the pericentromere, and have a conserved, yet poorly understood, function in biorientation (*Indjeian et al., 2005*; *Huang et al., 2007*; *Kiburz et al., 2008*; *Gutiérrez-Caballero et al., 2012*). In fission yeast, frogs, and human cells, shugoshins enable biorientation, at least in part, through recruitment of the chromosome passenger complex (CPC) containing aurora B to the pericentromere (*Kawashima et al., 2007*; *Vanoosthuyse et al., 2007*; *Kelly et al., 2010*; *Tsukahara et al., 2010*; *Wang et al., 2010*; *Yamagishi et al., 2010*; *Rivera et al., 2012*). Shugoshins also have a more defined role in protecting pericentromeric cohesin from premature loss during meiosis and mammalian mitosis; a function attributed to the recruitment of a specific form of the protein phosphatase 2A (PP2A) to the pericentromere (*Katis et al., 2004*; *Kitajima et al., 2004, 2006*; *Marston et al., 2004*; *Rabitsch et al., 2004*; *Riedel et al., 2006*; *Tang et al., 2006*; *Xu et al., 2009*).

**eLife digest** When a cell divides to create two new daughter cells, it must produce a copy of each of its chromosomes. It is important that each daughter cell gets just one copy of each chromosome. If an error occurs and one cell gets two copies of a single chromosome, it can lead to cancer or birth defects. Fortunately, there are multiple checks to ensure that this does not happen.

During cell division the chromosomes line up in a way that increases the likelihood that each daughter cell will have one copy of each chromosome. After this process—which is called biorientation—is completed, microtubules pull the chromosomes to opposite ends of the cell, which then divides.

Proteins called shugoshin proteins are known to be involved in biorientation in many organisms. These proteins are found in a region called the pericentromere, which surrounds the area on the chromosomes that the microtubules attach to, but the details of their involvement in biorientation are not fully understood. Now Verzijlbergen et al. have exploited sophisticated genetic techniques in yeast to explore how shugoshin proteins work.

These experiments showed that the shugoshin protein helps to recruit condensin—a protein that keeps the DNA organized within the chromosome—to the pericentromere to assist with biorientation. It also keeps aurora B kinase—one of the enzymes that helps to correct errors during cell division—in the pericentromere when a microtubule attaches to the wrong chromosome. These results help us understand how a 'hub' in the pericentromere ensures biorientation. The next challenge will be to understand how this hub is disassembled after biorientation to allow error-free cell division to proceed. As shugoshins have been found to be damaged in some cancers, understanding the workings of this hub could also shed new light on how they contribute to disease.

Though fundamental for accurate chromosome segregation, the role of shugoshin in biorientation has remained unclear. Budding yeast has a single shugoshin, Sgo1, which protects pericentromeric cohesin during meiosis but does not regulate cohesion during mitosis (*Katis et al., 2004*; *Kitajima et al., 2004*; *Marston et al., 2004*; *Indjeian et al., 2005*; *Kiburz et al., 2008*). We have exploited this system to investigate the sister kinetochore biorientation function of Sgo1, independently of effects on cohesion. Our analysis leads us to the unanticipated discovery that shugoshin collaborates with the chromosome-organising complex, condensin, in chromosome biorientation. Moreover, we provide the first molecular insight into how sister kinetochores are biased towards capture by microtubules from opposite, rather than the same, pole.

## Results

### Shugoshin is important for chromosome biorientation

To investigate the role of Sgo1 in biorientation we analyzed *sgo1* null cells (*sgo1Δ)* together with three missense mutants: *sgo1-100, sgo1-700* and *sgo1-3A*. The *sgo1-3A* mutant was engineered to disrupt the binding site for PP2A-Rts1 (*Xu et al., 2009*) whereas *sgo1-100* and *sgo1-700* were isolated in a screen due to their inability to sense a lack of tension (*Indjeian et al., 2005*). All three mutants and *sgo1Δ* cells have previously been reported to affect biorientation after microtubule perturbation (*Fernius and Hardwick, 2007*; *Indjeian et al., 2005*; *Indjeian and Murray, 2007*; *Xu et al., 2009*). The Sgo1-3A protein retains its pericentromeric localization (*Xu et al., 2009*; *Figure 1A*). Though the kinetics of cell cycle entry in *sgo1-100* and *sgo1-700* mutants is similar to that of wild-type cells (*Figure 1—figure supplement 1*), Sgo1-100 and Sgo1-700 show only residual initial centromeric recruitment (*Figure 1B*) and are absent from the pericentromeres of cells arrested in mitosis with microtubule-depolymerizing drugs (*Figure 1A*). We compared the ability of these Sgo1 mutants to establish bipolar attachments at metaphase after entering the cell cycle in the absence of microtubules. We used strains with spindle pole bodies (SPBs) labeled with tdTomato (*SPC42*-tdTomato), the centromere of chromosome IV labeled with GFP (*CEN4*-GFP) and with *CDC20* under control of the methionione-repressible promoter (*pMET-CDC20*), to enable metaphase arrest by addition of methionine. All strains were released from a G1 arrest into nocodazole- and methionine-containing medium to depolymerize microtubules and induce a metaphase arrest before nocodazole was washed

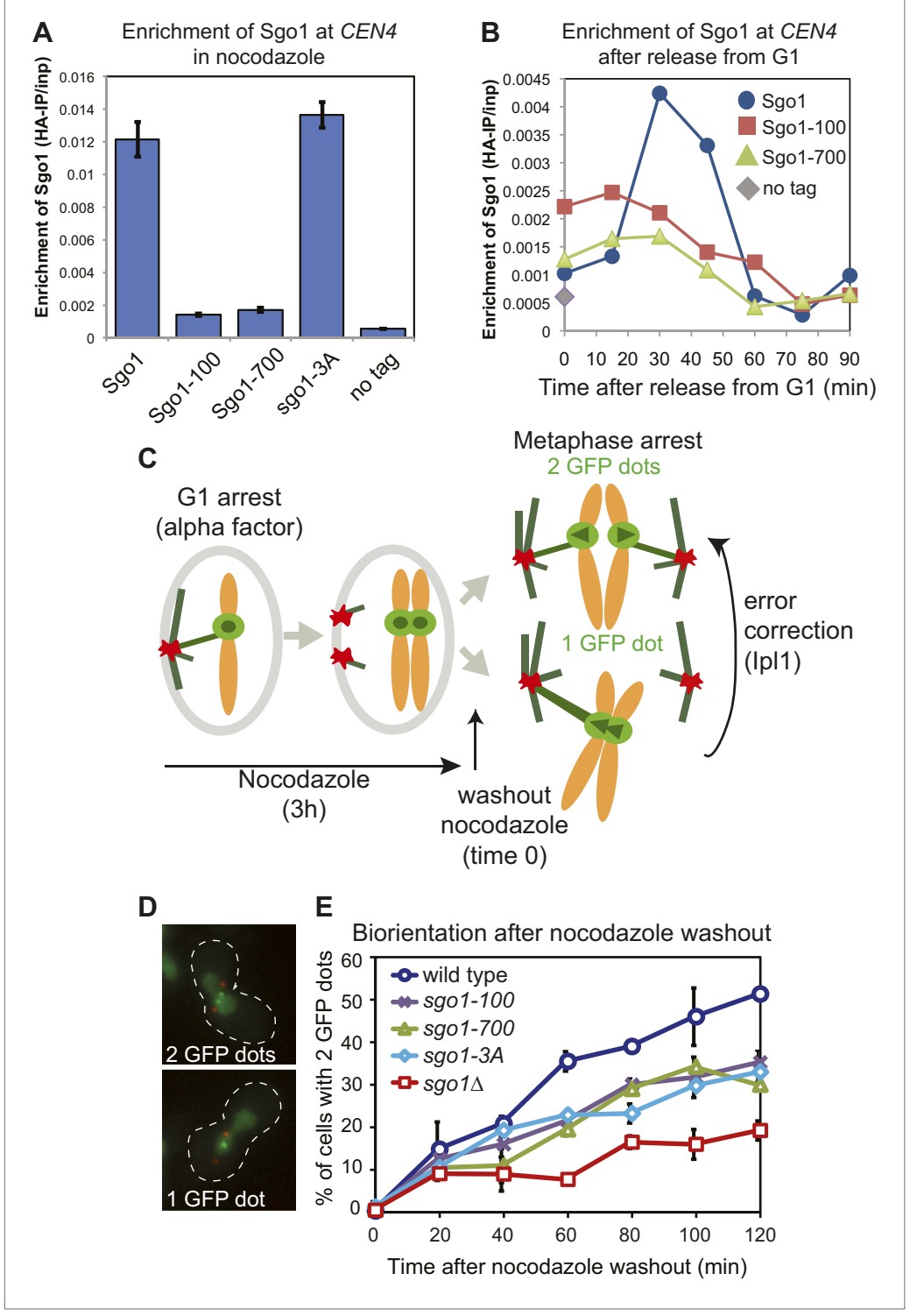

**Figure 1**. Sgo1 alleles that affect biorientation. (**A**) Sgo1-3A, but not Sgo1-100 or Sgo1-700 are maintained at the centromere in cells arrested in mitosis by treatment with nocodazole. Cells carrying *SGO1-6HA* (AM906), *SGO1-100-6HA* (AM6956), *SGO1-700-6HA* (AM6957), *SGO1-3A-6HA* (AM10011) and a no tag control (AM1176) were arrested in mitosis by treating with nocodazole for 3 hr. Cells were harvested for anti-HA ChIP and the levels of each Sgo1 protein at *CEN4* were analyzed by qPCR. The mean of three independent experiments is shown with error bars representing standard error. (**B**) Sgo1-100 and Sgo1-700 proteins are initially recruited to centromeres but fail to be maintained there. Strains as in (**A**) were arrested in G1 by treatment with alpha factor. Samples were
*Figure 1. Continued on next page*

*Figure 1. Continued*

extracted for analysis by anti-HA ChIP at 15 min intervals after release from G1. The levels of Sgo1-6HA at *CEN4* at the indicated times after release from G1 are shown for a representative experiment. (**C–E**) *SGO1* mutants are impaired in biorientation. Wild-type (AM4643), *sgo1-100* (AM8924), *sgo1-700* (AM8925), *sgo1-3A* (AM8923) and *sgo1Δ* (AM6117) cells carrying SPB (Spc42-tdTomato) and *CEN4* (*CEN4-GFP*) markers were released from a G1 arrest into medium containing nocodazole (to depolymerize microtubules) and methionine (to deplete *CDC20*). After 3 hr, nocodazole was washed out, and the number of GFP dots was scored in the metaphase-arrested cells as shown in the schematic diagram (**C**). (**D**) Representative images of cells with one and two GFP dots are shown. (**E**) The percentage of visibly separated centromeres was determined at the indicated intervals after nocodazole washout (*t* = 0). Error bars indicate range (n = 2).

The following figure supplements are available for figure 1:

**Figure supplement 1**. The *sgo1-100* and *sgo1-700* mutations do not affect the timing of cell cycle entry.

out to allow metaphase spindle formation (**Figure 1C**). Under these conditions, initial kinetochore-microtubule attachments are frequently erroneous because they occur before spindle pole bodies have migrated apart, leading to a strong reliance on the error correction process driven by Aurora B. We measured the efficiency of biorientation by scoring the splitting of *CEN4-GFP* signals once metaphase spindles reform after nocodazole washout. The *sgo1-100*, *sgo1-700* and *sgo1-3A* mutants showed a similar delay and lower maximum level of biorientation that was not as pronounced as in *sgo1Δ* cells (**Figure 1D,E**).

## Sgo1 does not promote biorientation through PP2A-Rts1 recruitment to centromeres

The *sgo1-3A* mutation disrupts the interaction between Sgo1 and PP2A-Rts1 (**Figure 2A**), which is important for the protection of centromeric cohesion during meiosis (**Xu et al., 2009**). Although the cohesin complex is properly associated with chromosomes in *sgo1Δ* cells during mitosis and cohesion is not affected (**Indjeian et al., 2005**; **Kiburz et al., 2005**; see below), PP2A-Rts1 could perform additional functions in biorientation. Rts1 enrichment at the centromere during metaphase is virtually abolished in *sgo1Δ*, *sgo1-3A* and *sgo1-100*, and modestly reduced in *sgo1-700* cells (**Figure 2B**), even though Sgo1-100 and Sgo1-700 proteins retain the ability to associate with Rts1 (**Figure 2A**). However, the biorientation defect of the *sgo1* mutants cannot be caused by a failure to recruit Rts1 to centromeres because *rts1Δ* cells achieved biorientation with indistinguishable efficiency to wild-type cells (**Figure 2C**). Therefore, PP2A-Rts1 is not required for sister kinetochore biorientation and the *sgo1-3A* mutation must disrupt functions of Sgo1 other than its association with PP2A-Rts1.

## Sgo1 ensures the maintenance of aurora B at centromeres

In other systems, shugoshins are known to affect the association of the chromosomal passenger complex (CPC) containing aurora B kinase with centromeres (**Kawashima et al., 2007**; **Vanoosthuyse et al., 2007**; **Yu and Koshland, 2007**; **Kelly et al., 2010**; **Tsukahara et al., 2010**; **Wang et al., 2010**; **Yamagishi et al., 2010**; **Rivera et al., 2012**). Budding yeast Sgo1 similarly associates with aurora B (called Ipl1) (**Figure 3A**). Conditional inactivation of Sgo1 using the auxin-inducible degron (aid) system (**Nishimura et al., 2009**; **Figure 3B**) revealed that Sgo1 is not required for the initial recruitment of Ipl1 to centromeres but is important for its maintenance (**Figure 3C,D**). Indeed, in Sgo1-aid cells arrested in metaphase by treatment with nocodazole, Ipl1 was absent from *CEN4* (**Figure 3D**). Early Ipl1 centromere localization also does not require Alk1 and Alk2 (**Figure 3—figure supplement 1**), the homologs of Haspin kinase, which is important for centromeric CPC localization in fission yeast and mammals (**Kelly et al., 2010**; **Wang et al., 2010**; **Yamagishi et al., 2010**). The recruitment of Ipl1 early in the cell cycle may instead be due to association with the Ndc10/Cbf3 kinetochore protein that is known to recruit the CPC to centromeres (**Yoon and Carbon, 1999**; **Cho and Harrison, 2012**). Sgo1-independent Ipl1 localization early in the cell cycle (**Figure 3C**) can explain why Ipl1, but not Sgo1, is essential for biorientation in an unperturbed cell cycle, though Ipl1 inhibition and deletion of *SGO1* similarly impair biorientation after microtubule depolymerization (**Figure 1E**, **Figure 3—figure supplement 2**; **Biggins et al., 1999**; **Tanaka et al., 2002**; **Indjeian et al., 2005**; **Indjeian and Murray, 2007**). The Ipl1-Sgo1

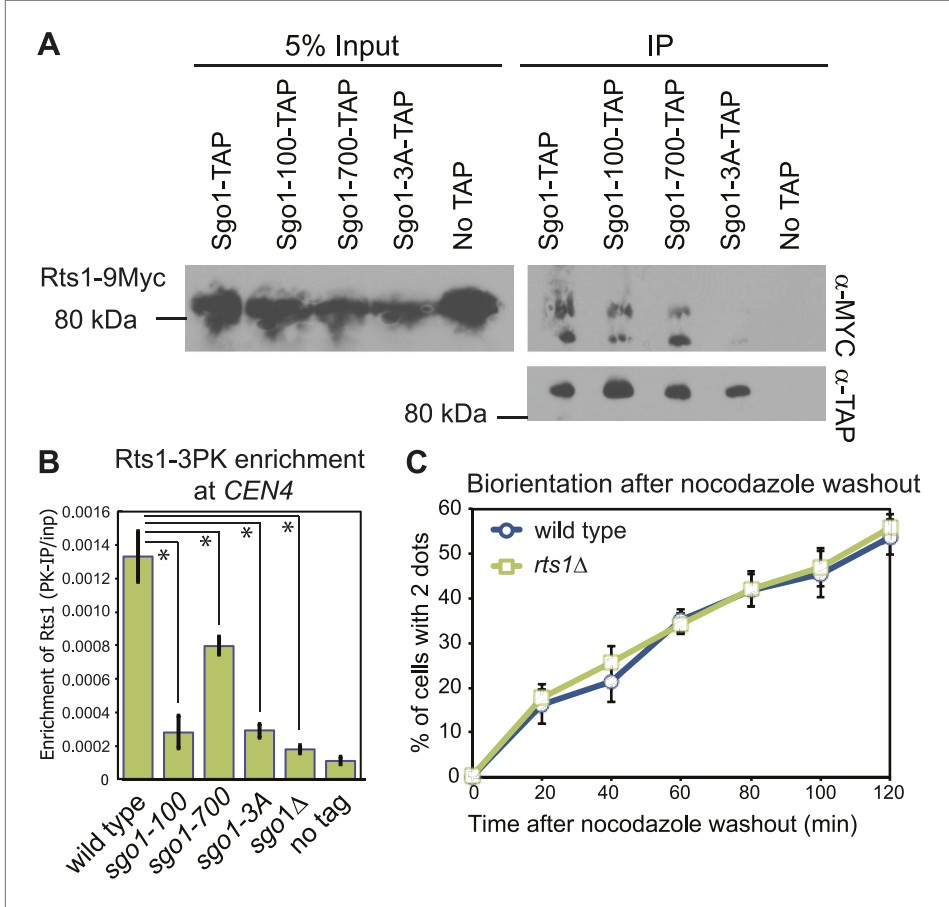

**Figure 2**. PP2A-Rts1 recruitment to the centromere by Sgo1 is not required for biorientation. (**A**) Sgo1-100 and Sgo1-700, but not Sgo1-3A, associate with Rts1. Cells carrying *RTS1-9MYC* and *SGO1-SZZ(TAP)* (AM9144), *sgo1-100-SZZ(TAP)* (AM9272), *sgo1-700-SZZ(TAP)* (AM9142), *sgo1-3A-SZZ(TAP)* (AM9145) or no TAP (AM4721) were arrested in nocodazole for 2 hr and treated with the cross-linking reagent dithiobis(succunimidylpropionate) (DSP) before extract preparation as described in 'Materials and methods'. Extracts were incubated with IgG-coupled beads and immunoprecipitates analyzed with the indicated antibodies. (**B**) Sgo1 mutants affect the centromeric localization of Rts1. Wild-type (AM8895), *sgo1-100* (AM9439), *sgo1-700* (AM9323), *sgo1-3A* (AM9293) and *sgo1Δ* (AM9624) cells carrying *RTS1-3PK*, as well as a no tag control (AM1176), were treated with nocodazole for 3 hr before harvesting for anti-PK ChIP. The mean level of Rts1-3PK enrichment at *CEN4* from three experimental repeats, determined by qPCR, is shown with bars indicating standard error (*p<0.05, paired *t* test). (**C**) Sister kinetochore biorientation after microtubule depolymerization was measured in wild-type (AM4643) and *rts1Δ* (AM5823) cells as in *Figure 1* (**C**). The mean of three experimental repeats with error bars representing standard deviation are shown.

interaction (*Figure 3A*) and centromeric localization of Ipl1 (*Figure 3E*) were also similarly decreased in nocodazole-treated *sgo1-100*, *sgo1-700* and *sgo1-3A* mutants, which likely contributes to the biorientation defects of these mutants.

## Sgo1 associates with condensin and recruits it to the pericentromere

As an unbiased approach to isolate binding partners that might contribute to biorientation we purified Sgo1 from cycling cells or cells arrested in mitosis by microtubule perturbation (using a cold-sensitive tubulin mutant; *tub2-401* [*Huffaker et al., 1988*]). To increase the probability of capturing transient interactions, we pre-treated cells with the cross-linking agent dithiobis(succunimidylpropionate) (DSP) before preparing extracts and immune-precipitating Sgo1-TAP. Associated proteins were identified by mass spectrometry (*Figure 4A,B*; *Supplementary file 1*). Although subunits of PP2A co-purified with Sgo1, we did not detect peptides of the CPC. Interestingly, we identified four out of five subunits of the

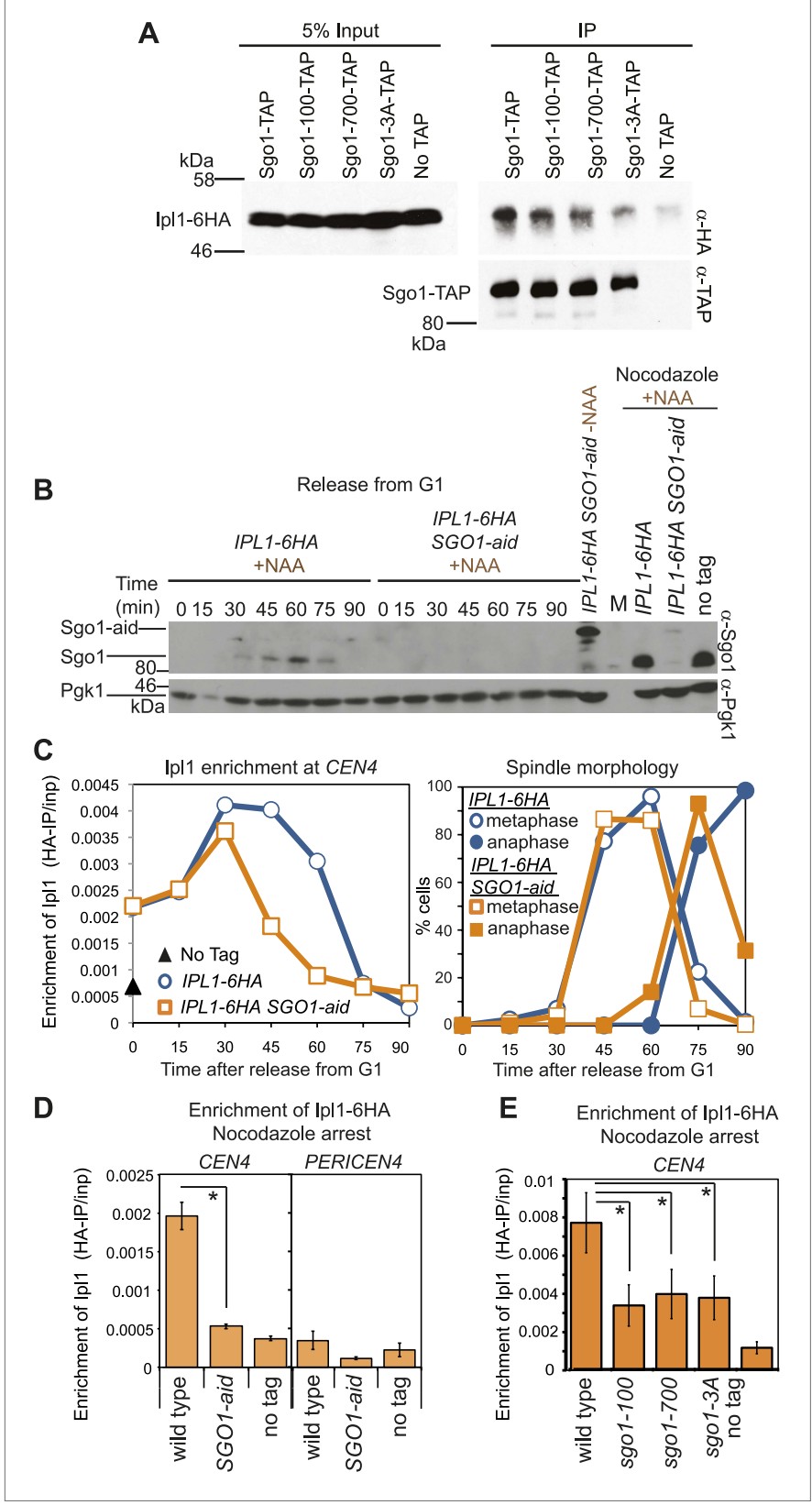

**Figure 3**. Sgo1 is required for the maintenance of Ipl1 at centromeres, but is dispensable for its initial recruitment. (**A**) Ipl1/aurora B co-immunoprecipitates with Sgo1. Cells producing SZZ(TAP)-tagged Sgo1 (AM8975), Sgo1-100 (AM8971), Sgo1-700 (AM8969), Sgo1-3A (AM8973) or no TAP (AM3513) and carrying *IPL1-6HA* were treated with

*Figure 3. Continued*

nocodazole for 2 hr before cross-linking with DSP. Extracts were prepared as described in 'Materials and methods', incubated with IgG-coupled beads and immunoprecipitates were analyzed by immunoblot using the indicated antibodies. (**B**) Degradation of Sgo1 using the auxin-inducible degron system. Representative anti-Sgo1 immunoblot for the experiments in (**C** and **D**) showing that NAA treatment leads to Sgo1 degradation. Anti-Pgk1 immunoblot is shown as a loading control. See below for experimental conditions. (**C**) Ipl1 is initially recruited to centromeres in the absence of Sgo1. Wild-type (AM3513) and *SGO1-aid* (AM9619) cells carrying *IPL1-6HA* were released from a G1 block in the presence of auxin (NAA) and samples harvested at 15 min intervals for measurement of Ipl1-6HA levels by anti-HA ChIP-qPCR. Also shown is a G1 sample from cells lacking *IPL1-6HA* (AM1176; no tag). The percentages of metaphase and anaphase spindles after anti-tubulin immunoflurescence were scored as a marker of cell cycle progression and anti-Sgo1 immunoblot confirmed Sgo1-aid degradation (shown in **B**). A representative experiment is shown from a total of three repeats. (**D**) Wild-type (AM3513) and *SGO1-aid* (AM9619) cells carrying *IPL1-6HA* together with a no tag control were arrested in G1 by alpha factor treatment and then released into medium containing NAA and nocodazole for 3 hr before harvesting for ChIP. Levels of Ipl1-6HA were determined at *CEN4* and a pericentromeric site (*PERICEN4*) by qPCR and the mean of three experimental repeats is shown with bars representing standard error (*p<0.05, paired *t* test). (**E**) Ipl1-6HA levels at *CEN4* measured by anti-HA ChIP-qPCR in wild type (AM3513), *sgo1-100* (AM9090), *sgo1-700* (AM9082) and *sgo1-3A* (AM9076) after treating directly with nocodazole for 3 hr are shown, together with a no tag (AM1176) control, treated in the same way. The mean of three independent repeats is shown with bars representing standard error (*p<0.05, paired *t* test). Note that levels of Ipl1-6HA at *CEN4* were consistently higher in experiments where cells were directly treated with nocodazole, compared to those treated upon release from G1 (compare **E** with **D**). Presumably those cells in the population that are already in mitosis upon nocodazole addition experience an extended arrest during which Ipl1 is continually recruited.

The following figure supplements are available for figure 3:

**Figure supplement 1**. The Haspin homologs, Alk1 and Alk2, are not required for the initial recruitment of Ipl1 to centromeres.

**Figure supplement 2**. Defective biorientation in *ipl1-as* mutants.

---

condensin complex co-purifying with Sgo1 (*Figure 4A,B*; *Supplementary file 1*). Co-immunoprecipitation of the Ycs4 and Brn1 subunits of condensin (*Figure 4C*) with Sgo1-TAP confirmed the Sgo1-condensin interaction (*Figure 4D,E*). We confirmed that the Sgo1-Ycs4 interaction is not dependent on either DNA or the pre-treatment of cells with cross-linking agent (*Figure 4—figure supplement 1*). This suggests that Sgo1 and condensin form a complex independently of their association with the pericentromeric chromatin. Therefore, Sgo1 associates with three protein complexes during mitosis: PP2A, CPC, and condensin.

Condensin complexes structurally organize chromosomes and enable their efficient segregation, though how they do so remains unclear (*Cuylen and Haering, 2011*; *Hirano, 2012*). In budding yeast, condensin is most highly enriched in the rDNA and at each pericentromere (*D'Ambrosio et al., 2008*). Condensin recruitment to the rDNA depends on monopolin (Csm1/Lrs4) (*D'Ambrosio et al., 2008*). Fission yeast monopolin (Pcs1/Mde4) recruits condensin to centromeres where it prevents merotely (attachment of a single kinetochore to microtubules from both poles) (*Gregan et al., 2007*; *Tada et al., 2011*), unlike budding yeast condensin which is recruited to centromeres independently of monopolin subunit Lrs4 (*Brito et al., 2010*). How condensin is recruited to the pericentromere remains unknown. To test whether the pericentromeric localization of condensin depends on Sgo1 we examined the association of the Brn1 condensin subunit genome wide using chromatin immunoprecipitation followed by high throughput sequencing (ChIP-seq) in wild-type and *sgo1Δ* cells arrested in mitosis by treatment with nocodazole. Although the pattern of reads along chromosome arms and mapping to the rDNA was similar in wild-type and *sgo1Δ* cells (*Figure 4F,G*), we observed a clear reduction in pericentromeric levels of Brn1 in *sgo1Δ* cells, although peaks of variable height remained at some, but not all core centromeres (*Figure 4H*, *Figure 4—figure supplement 2, 3*). Consideration of all 16 centromeres collectively revealed that in wild-type cells, condensin is enriched on average throughout an approximately 15 kb domain on either side of the centromere and that this enrichment is lost in *sgo1Δ* cells (*Figure 4H*). We conclude that Sgo1 is required for condensin association throughout the pericentromere.

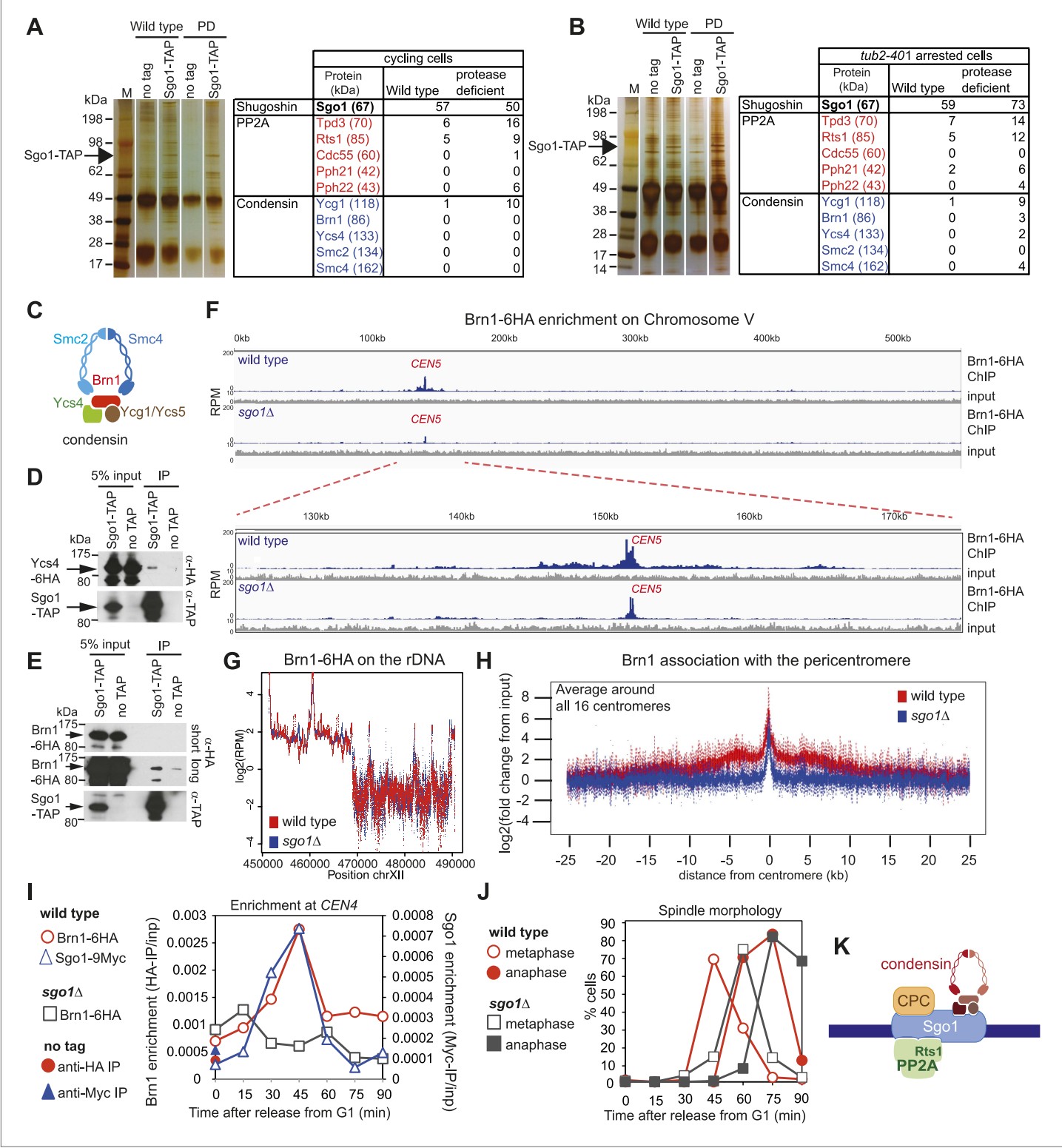

**Figure 4**. Sgo1 interacts with condensin and recruits it to the pericentromere. (**A** and **B**) Condensin and PP2A co-purify with Sgo1. Sgo1 was purified from wild-type or protease-deficient cells that were (**A**) cycling or (**B**) arrested in mitosis using the cold-sensitive tubulin allele *tub2-401* as described in 'Materials and methods'. Comparable strains lacking TAP were used as a control for non-specific association with the beads. All cells were treated with the cross-linker, DSP, before harvesting and preparing extracts as described in 'Materials and methods'. Extracts were incubated with IgG-coupled beads and immunoprecipitates were visualized on silver-stained SDS-PAGE gels. The table shows the number of peptides of subunits of
*Figure 4. Continued on next page*

*Figure 4. Continued*

the PP2A and condensin complexes that were identified in the Sgo1-TAP purifications after mass spectrometry. The full list of identified proteins is given in *Supplementary file 1*. Strains used in (**A**) were AM7509 (*SGO1-SZZ(TAP)*), AM1176 (no tag), AM8226 (protease-deficient, *SGO1-SZZ(TAP)*) and AM8184 (protease-deficient, no tag). Strains used in (**B**) were AM8456 (*tub2-401 SGO1-SZZ(TAP)*), AM2730 (*tub2-401,* no tag), AM8455 (protease-deficient, *tub2-401 SGO1-SZZ(TAP)*) and AM8259 (protease-deficient, *tub401* no tag. (**C**) A schematic diagram illustrating the composition of budding yeast condensin is shown. (**D** and **E**) Cells carrying either *YCS4-6HA* (AM9138) (**D**) or *BRN1-6HA* (AM9266) (**E**) and *SGO1-SZZ(TAP)* or no TAP (AM5705 and AM5708) were arrested in nocodazole for 2 hr before treating with DSP. Extracts were incubated with IgG-coupled beads and immunoprecipitates were analyzed with the indicated antibodies by immunoblot. (**F**–**H**) Sgo1 is required for Brn1 association with the pericentromere. The genome-wide localization of Brn1-6HA was determined in wild-type (AM5708) and *sgo1Δ* (AM8834) cells by anti-HA ChIP followed by high throughput sequencing (ChIP-Seq) after arresting in mitosis by treating with nocodazole for 3 hr. (**F**) Brn1 enrichment along chromosome V along with a magnification of a 50 kb region including the centromere is shown. The number of reads at each position was normalized to the total number of reads for each sample (RPM: reads per million) and shown in the Integrated Genome Viewer from the Broad Institute (*Robinson et al., 2011*). (**G**) The number of reads at coordinates corresponding to the rDNA region on chromosome XII is shown for wild-type and *sgo1Δ* anti-HA ChIP samples normalized to the total number of reads for each sample. Brn1 enrichment at the rDNA is similar in wild-type and *sgo1Δ* cells. (**H**) Brn1 enrichment in a 50 kb domain surrounding all 16 budding yeast centromeres is shown for wild-type and *sgo1Δ* cells. For both wild type and *sgo1Δ*, the ratio of the local maximum in a 100 bp window for ChIP sample/input is calculated at the indicated distance from the centromere for all 16 chromosomes. Box plot of maximum count value for 100 bp windows for 25 kb on both sides of each centromere is shown to give a composite view of all 16 pericentromeres. (**I** and **J**) Recruitment of Brn1 to centromeres occurs coincident with, and is dependent on, Sgo1. Wild-type cells carrying *SGO1-9MYC* and *BRN1-6HA* (AM9622) as well as *sgo1Δ* cells (AM8834) carrying *BRN1-6HA* were arrested in G1 using alpha factor. Samples were extracted at 15 min intervals after release from G1 for anti-HA and anti-Myc ChIP and tubulin immunofluorescence. (**I**) The levels of Brn1-6HA and Sgo1-9Myc at *CEN4* were measured at the indicated timepoints by anti-HA and anti-Myc ChIP-qPCR, respectively. Also shown is a G1 sample from cells lacking *BRN1-6HA* (no tag; AM1176). (**J**) The percentages of metaphase and anaphase spindles after anti-tubulin immunofluorescence were scored as a marker of cell cycle progression. Shown is a representative experiment from three repeats. (**K**) Schematic diagram illustrating the protein complexes (PP2A, condensin, CPC) recruited to the pericentromere by shugoshin.

The following figure supplements are available for figure 4:

**Figure supplement 1**. The Sgo1-condensin interaction is not dependent on DNA or treatment with the cross-linking agent, DSP.

**Figure supplement 2**. Removal of PCR duplicates does not alter the conclusion that Sgo1 is important for Brn1 enrichment in the pericentromere.

**Figure supplement 3**. Brn1 is reduced around all 16 individual centromeres in *sgo1Δ* cells.

Sgo1 is absent in G1 and produced only upon cell cycle entry (*Marston et al., 2004*). Although condensin is present in G1 cells and localized to the nucleolus, it begins to co-localize with kinetochores only upon cell cycle entry (*Bachellier-Bassi et al., 2008*). We found that recruitment of condensin to a centromere-proximal site occurs coincidently with, and depends on, Sgo1 (*Figure 4I,J*). Therefore, in addition to controlling the centromere localization of PP2A-Rts1 and the CPC, Sgo1 recruits condensin to the pericentromere (*Figure 4K*).

## Hierarchical assembly of pericentromeric factors

Like condensin and shugoshin (*D'Ambrosio et al., 2008*; *Kiburz et al., 2005*; *Figure 4*), cohesin is highly enriched throughout the pericentromere (*Glynn et al., 2004*; *Lengronne et al., 2004*; *Weber et al., 2004*). What is the relationship between cohesin, condensin, and shugoshin? Although shugoshins play important roles in regulating the timing of cohesion loss during meiosis and mammalian mitosis (see *Clift and Marston, 2011*; *Gutiérrez-Caballero et al., 2012* for reviews), this is not the case in budding yeast mitosis. Budding yeast *sgo1* mutants are not defective in cohesion (*Indjeian and Murray, 2005*; *Marston et al., 2004*) and cohesin is normally localized to chromosomes (*Kiburz et al., 2005*). We confirmed the proper association of cohesin in *sgo1Δ* cells arrested in mitosis by ChIP-Seq of its HA-tagged Scc1 subunit (*Figure 5A,B*). The profile of Scc1 association along chromosome V (*Figure 5A*) and surrounding all 16 budding yeast centromeres in *sgo1Δ* cells (*Figure 5B*, *Figure 5—figure supplement 1*) is indistinguishable from that of wild-type cells. ChIP-qPCR analysis confirmed that the levels of Scc1 cohesin subunit are similar at two tested centromeres in wild-type and *sgo1Δ* cells (*Figure 5—figure supplement 2*). We conclude that Sgo1 is not required for cohesin localization at centromeres, pericentromeres or along chromosomes.

Conversely, we found that cohesin, and factors required for its loading, are required for the proper association of Sgo1 with the centromere and pericentromere (*Figure 5*, *Figure 5—figure supplements 3 and 4*). Depletion of the Scc1 subunit of cohesin led to a great reduction in the

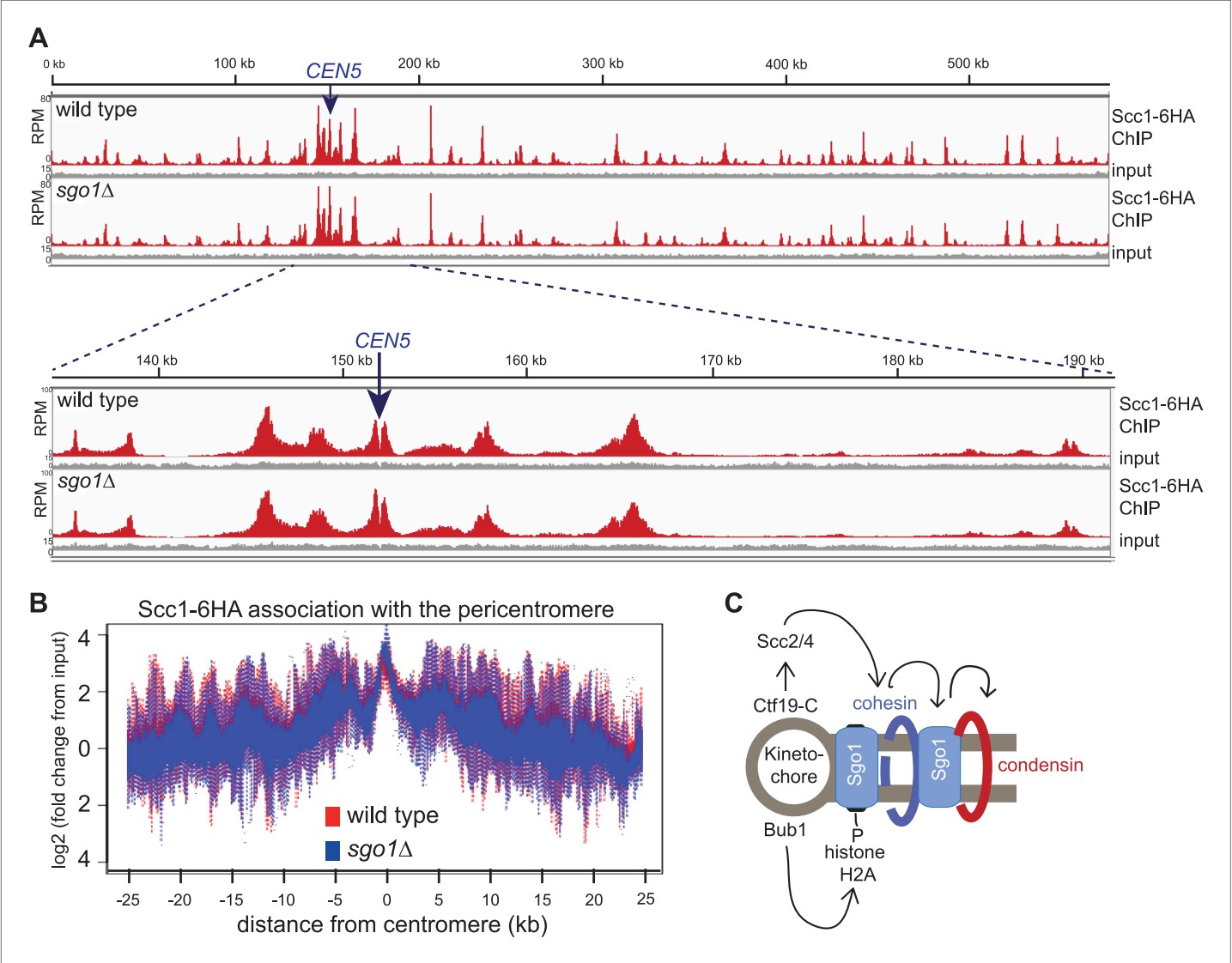

**Figure 5**. Sgo1 is not required for cohesin association with chromosomes. Wild-type (AM1145) and *sgo1Δ* (AM1474) cells carrying *SCC1-6HA* were arrested in mitosis by treatment with nocodazole for 3 hr. Samples were harvested, anti-HA ChIP was performed and both input and IP samples were sequenced for both strains. (**A**) Scc1-6HA enrichment along chromosome V along with a magnification of a 50 kb region including the centromere is shown. The number of reads at each position were normalized to the total number of reads for each sample and displayed using the Integrated Genome Viewer from the Broad Institute (*Robinson et al., 2011*). (**B**) Box plot of maximum count value for 100 bp windows for 25 kb on both sides of each centromere is shown to give a composite view of all 16 pericentromeres. All reads are included. (**C**) Schematic diagram indicating hierarchy of factors required for condensin association with the pericentromere.

The following figure supplements are available for figure 5:

**Figure supplement 1**. Scc1 association with all 16 centromeres is unaffected by *SGO1* deletion.

**Figure supplement 2**. ChIP-qPCR analysis showing Scc1-6HA levels at the centromere and pericentromere.

**Figure supplement 3**. Cohesin is required for normal Sgo1 association with the pericentromere.

**Figure supplement 4**. Cohesin loading factors, but not monopolin, are important for proper condensin association with the centromere.

pericentromeric levels of Sgo1 with only low levels remaining at the centromere itself (*Figure 5—figure supplement 3*). These findings suggest that cohesin promotes Sgo1 association with the pericentromere, which, in turn, recruits condensin, implying an indirect role for cohesin in localizing condensin (through Sgo1) (*Figure 5C*). Consistent with this idea, proper Brn1 association with the centromere in mitosis requires both the Scc2 protein that is required for cohesin loading onto chromosomes and subunits of the kinetochore (Iml3/Chl4) that target Scc2 to centromeres (*Ciosk et al., 2000*; *Fernius et al., 2013*; *D'Ambrosio et al., 2008*; *Ng et al., 2009*; *Figure 5—figure supplement 4*), However, unlike in fission yeast, the monopolin subunit Lrs4 is not required for Brn1 association with the centromere (*Brito et al., 2010*; *Figure 5—figure supplement 4*). Overall, we suggest a hierarchy of assembly in which the combined effects of Bub1 kinase and cohesin concentrate shugoshin in the pericentromere, which in turn recruits condensin (*Fernius and Hardwick, 2007*; *Kawashima et al., 2010*; *Yamagishi et al., 2010*; *2008*) (*Figure 5C*).

## Sgo1 is sufficient for condensin recruitment

Next, we asked whether Sgo1 was sufficient to recruit condensin to chromosomes. Overproduction of Sgo1, which is known to enable its association with chromosome arms and delay cells in metaphase (*Clift et al., 2009*; *Figure 6A*) led to increased levels of condensin at centromere, pericentromere and chromosome arm sites (*Figure 6B*). *SGO1* overexpression also increased Brn1 association with the centromere and pericentromere in cells arrested in mitosis by nocodazole treatment (*Figure 6—figure supplement 1*), indicating that increased enrichment of Brn1 in *SGO1*-overexpressing cells was not purely a consequence of the metaphase arrest. As a more direct test of the ability of Sgo1 to bring condensin to chromosomes, we produced a Sgo1-GFP-TetR fusion protein in cells carrying *tetO* repeats integrated on a chromosomal arm. In the absence, but not the presence of doxycycline, Sgo1-GFP-TetR is expected to bind to the ectopic site and recruit its binding partners (*Figure 6C*). Indeed we found that tethered Sgo1-GFP-TetR efficiently recruited the condensin subunit Brn1, the PP2A subunit, Rts1, and to a lesser extent the CPC subunit, Ipl1 to a site directly adjacent to the *tetOs* (*Figure 6D,F*; ~50 bp R ectopic site), although centromeric levels were not affected (*Figure 6E,G*). The recruitment of these proteins to a site ~800 bp to the left of the tethering site was much less efficient (*Figure 6D,F*), suggesting that recruitment occurs through direct binding to Sgo1, rather than an effect of Sgo1 on the surrounding chromatin. Taken together, these results show that Sgo1 is both necessary and sufficient for condensin recruitment.

## Pericentromeric condensin contributes to error correction, independently of aurora B recruitment

Biorientation is achieved both because of a bias for sister kinetochores to be captured by microtubules from opposite poles and owing to error correction which destabilizes mono-oriented kinetochores, allowing a further opportunity for biorientation to occur (*Tanaka, 2010*). We tested the requirement of condensin for error correction using a conditional degron version of its Ycs5 subunit by monitoring *CEN4-GFP* separation in metaphase-arrested cells after microtubule depolymerization (*Figure 1C*). In cells where degradation of condensin's Ycs5 subunit (*YCS5-aid*) was induced, *CEN4-GFP* separation was delayed compared to wild-type cells, albeit not to the extent of *sgo1Δ* cells (*Figure 7A*), suggesting that condensin facilitates biorientation. To further test whether the error correction process operates normally we developed a live cell microfluidics assay to allow biorientation to be observed directly as microtubules were allowed to reform (*Figure 7B,C*; *Videos 1 and 2*). Overall, as expected, and consistent with a biorientation defect, the number of frames in which cells with split *CEN4-GFP* foci were observed was reduced in cells lacking *SGO1*, where *IPL1* was inhibited (*ipl1-as* with NAPP1) or Ycs5 was degraded (Ycs5-aid with NAA) compared to wild-type cells (*Figure 7D*). However, the distance between separated *CEN4-GFP* foci was comparable in all strains, suggesting that kinetochore-microtubule attachments, spindle tension, and cohesion are all functional and only the orientation of attachment is defective in *sgo1Δ*, *YCS5-aid* and *ipl1-as* cells (*Figure 7E*). Furthermore, consistent with a failure to properly biorient chromosomes, unseparated *CEN4-GFP* tended to be closer to the SPB in *sgo1Δ*, *YCS5-aid* and *ipl1*-as cells than in wild-type cells (*Figure 7—figure supplement 1*). We used the switch between one or two *CEN4-GFP* foci as a measure of kinetochore reorientation during error correction (*Figure 7F*). The average frequency of switching between 1 and 2 GFP foci was significantly reduced in *sgo1Δ* and *ipl1-as* cells, as expected (*Figure 7F*). We further observed a more modest reduction in switching in *YCS5-aid* cells, indicating that condensin contributes to

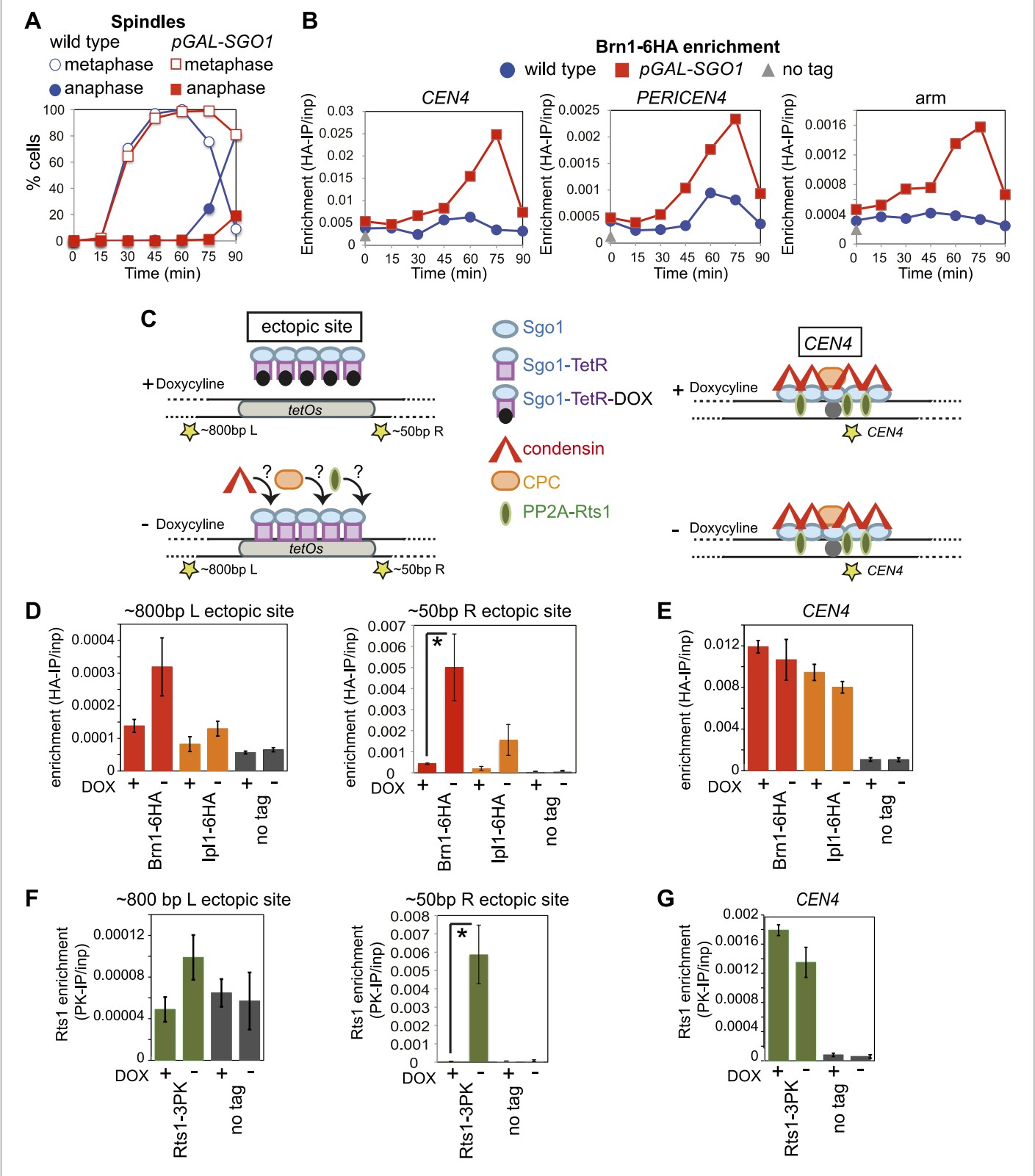

**Figure 6**. Sgo1 is sufficient for condensin recruitment. (**A** and **B**) Sgo1 overproduction leads to increased levels of Brn1 on chromosomes. Cells carrying *BRN1-6HA* and that were otherwise wild type (AM5708) or carrying *pGAL-SGO1* (AM10859) integrated an ectopic locus, were arrested in G1 in rich medium containing raffinose and adenine using alpha factor (YEP + R + A). After 2 hr 30 min, galactose (2%) was added to induce *SGO1* overexpression
*Figure 6. Continued on next page*

*Figure 6. Continued*

and 30 min later, cells were released from G1. Samples were collected at the indicated times after release from G1 for analysis of cell cycle progression by scoring spindle morphology after anti-tubulin immunofluorescence (**A**) or for measurement of Brn1 levels by anti-HA ChIP-qPCR (**B**). Sites analyzed were at *CEN4*, a pericentromeric site or a chromosomal arm site on chromosome IV. A representative experiment from a total of three independent repeats is shown. (**C–G**) Tethered Sgo1 at an ectopic site recruits Brn1, Rts1 and Ipl1. (**C**) Schematic diagram showing the expected effects of doxycycline at the ectopic site and at *CEN4*, as well as the locations of primer sets used for qPCR (yellow stars). Primer sets used were ~800 bp left of the tethering site, ~50 bp right of the tethering site and at *CEN4*. (**D–G**) Strains carrying Sgo1-TetR-GFP and *tetOs* integrated at the *HIS3* locus were arrested in nocodazole for 3 hr either in the presence (+DOX) or absence (−DOX) of doxycycline and harvested for ChIP-qPCR. (**D** and **E**) Anti-HA ChIP was performed on *SGO1-TetR-GFP HIS3::tetOs* strains carrying either *BRN1-6HA* (AM9847), *IPL1-6HA* (AM9940) or no tag (AM9655) and levels of Brn1-6HA and Ipl1-6HA were measured by qPCR at the indicated sites adjacent to the ectopic site (**D**) or at *CEN4* (**E**). (**F** and **G**) Anti-PK ChIP was performed on *SGO1-TetR-GFP HIS3::tetOs* strains carrying Rts1-3PK (AM9783) or no tag (AM9655) and levels of Rts1-3PK were measured by qPCR at the indicated sites adjacent to the ectopic site (**F**) or at *CEN4* (**G**). In (**D–G**), the mean of three or four experimental repeats is shown with bars representing standard error (*p<0.05, unpaired *t* test).

The following figure supplements are available for figure 6:

**Figure supplement 1**. *SGO1* overexpression in metaphase-arrested cells increases Brn1 association with the centromere.

proper error correction (*Figure 7F*). The role of condensin in error correction cannot be to localize Ipl1, as we found that Ipl1 maintenance at kinetochores requires Sgo1 (*Figure 3D*), but not Ycs5 (*Figure 7G*). Rather, we speculate that condensin shapes the pericentromere to place sister kinetochores in a rigid back-to-back orientation that provides the framework for tension-sensing.

## Shugoshin and pericentromeric condensin bias sister kinetochores to capture by microtubules from opposite poles

Although it has been recognized that sister kinetochores are intrinsically biased towards capture by microtubules from opposite poles (*Indjeian and Murray, 2007*), the factors required were not identified. The molecular basis of the bias towards sister kinetochore biorientation has therefore remained unknown. If our hypothesis that condensin creates a preferred pericentromeric framework upon which the error correction machinery can act is correct, we reasoned that condensin could also impose a bias on kinetochores to biorient. When SPBs are allowed to separate before microtubules attach to sister kinetochores, they tend to biorient normally, even when error correction is impaired (*Indjeian and Murray, 2007*). To test the requirement of shugoshin and condensin for sister kinetochore bias, we allowed cells to progress from G1 into the cell cycle for 1.5 hr to allow SPB separation, before treating the cells with nocodazole for 30 min. Subsequently, nocodazole was washed out and we simultaneously began filming (*Figure 8A*). We recorded the percentage of cells with separated SPBs at the start of filming that separated *CEN4-GFP* foci at least once during the observation period (approximately 30 min) (*Figure 8B,C*; *Videos 3 and 4*). Similar numbers of wild-type and *ipl1-as* cells achieved sister centromere separation, indicating that the error correction process is not required to bias sister kinetochores towards biorientation (*Figure 8B*). However, remarkably, the frequency of separated *CEN4-GFP* foci was reduced about two-fold in *sgo1Δ* and *YCS5-aid* cells, as compared to wild-type cells (*Figure 8B*). This indicates that in the absence of shugoshin or pericentromeric condensin, the bias to sister kinetochore biorientation is lost. We conclude that both shugoshin and condensin impose a bias on sister kinetochores to biorient and that this is independent of error correction by Aurora B (Ipl1).

## The bias towards sister kinetochore biorientation relies on the shugoshin-condensin interaction

Shugoshins are emerging as factors that define a hub at the pericentromere that integrates the functions of multiple protein complexes to ensure the accuracy of chromosome segregation (*Figure 9A*) (*Rattani et al., 2013*). To explore the relationship between shugoshin and its binding partners further, we asked whether the Sgo1-100, Sgo1-700 and Sgo1-3A proteins retain their interaction with condensin. We found that Sgo1-700 failed to associate with Brn1, whereas Sgo1-100 and Sgo1-3A retained Brn1 association (*Figure 9B*). Analysis of Brn1 association with representative centromere-proximal and chromosomal arm sites by ChIP-qPCR revealed that only the *sgo1-3A* mutant, and not the *sgo1-100* or *sgo1-700* mutants maintained Brn1 localization at the pericentromere in cells arrested in mitosis by

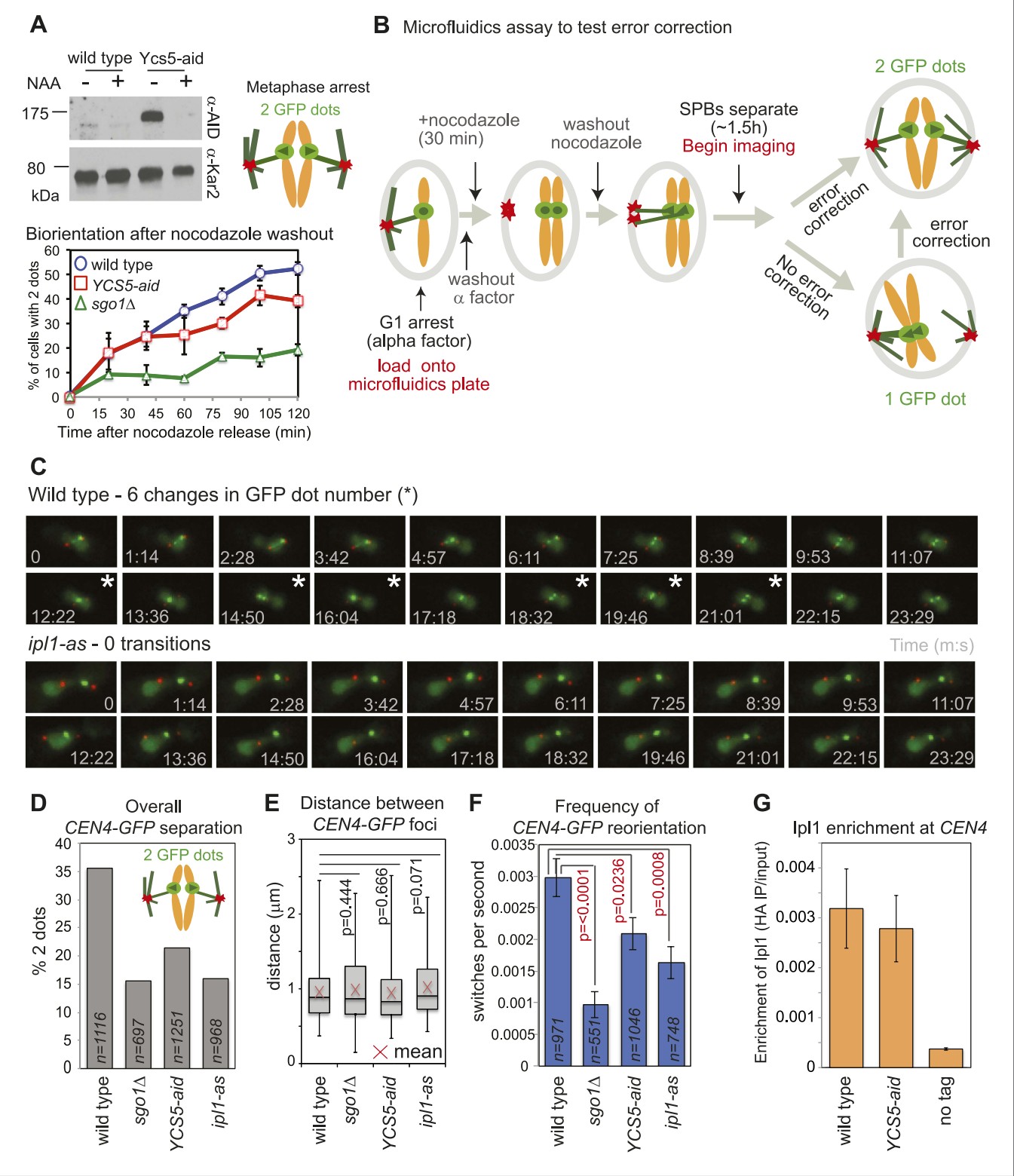

**Figure 7.** Condensin facilitates effective error correction. (**A**) Sister kinetochore biorientation is defective after nocodazole washout in cells lacking condensin. Strains carrying SPB (Spc42-tdTomato) and *CEN4* (*CEN4-GFP*) markers were released from a G1 arrest into nocodazole and arrested in metaphase by *CDC20* depletion. After 3 hr, nocodazole was washed out (*t* = 0) and *CEN4*-GFP separation scored at the indicated intervals as shown in *Figure 1C*. Error bars represent standard deviation (wild type and *YCS5-aid*; n = 3) or range (*sgo1Δ*; n = 2; reproduced from *Figure 1E*). A representative

*Figure 7. Continued*

anti-aid immunoblot is shown to confirm Ycs5 degradation upon NAA addition. Samples were taken in G1 (−NAA) and 120 min after release (+NAA). Anti-Kar2 immunoblot is shown as a loading control. (**B–F**) Condensin contributes to efficient error correction. (**B**) Scheme of the live single-cell microfluidics experiment. Wild-type (AM4643), *sgo1Δ* (AM6117), *YCS5-aid* (AM9038) and *ipl1-as* (AM10374) cells carrying *CEN4* (*CEN4-GFP*) and SPB markers (*SPC42-tdTomato*) were released from a G1 arrest into nocodazole, NAA and NAPP1 and arrested in metaphase by *CDC20* depletion. After 30 min, nocodazole was washed out. When the majority of cells had 2 SPBs (~1 hr 30 min later), we began imaging and a total of 21 frames were grabbed at approximately 74 s intervals. (**C**) Representative images for wild type and *ipl1-as* are shown. Numbers indicate time (s) each frame was grabbed and asterisks indicate a change in GFP dot number compared to the previous frame. (**D**) The overall percentage of separated *CEN4-GFP* foci was determined for cells with two visible SPBs from all frames combined. (**E**) The distance between *CEN4-GFP* foci was measured in cells with separated foci. Box boundaries represent the upper and lower quartiles, respectively. The red cross indicates the mean, the horizontal line indicates the median and error bars show the maximum and minimum values observed. n = 396 (wild type), 108 (*sgo1Δ*), 267 (*YCS5-aid*) and 154 (*ipl1-as*). (**F**) The observed frequency of switching between one and two GFP foci was calculated for cells with 2 SPBs. A student *t* test was used to obtain p values. (**G**) Ycs5 is not required for Ipl1 association with the centromere. The levels of 6HA-tagged Ipl1 in wild-type (AM3513) and *YCS5-aid* (AM10334) cells, grown in the presence of NAA and nocodazole for 3 hr, were measured at *CEN4* by anti-HA ChIP-qPCR and compared to a no tag control (AM1176). The mean of three independent experiments is shown with bars representing standard error. This is the same experiment as shown in *Figure 3D* and the wild-type data is reproduced for comparison.

The following figure supplements are available for figure 7:

**Figure supplement 1**. Deletion of *SGO1* impairs biorientation rather than centromere cohesion.

nocodazole treatment (*Figure 9C*). However, in both *sgo1-100* and *sgo1-700* mutant cells progressing from G1 into the cell cycle, Brn1-6HA retained a partial ability to associate with the centromere and pericentromere, though only in the case of the *sgo1-100* did this occur in a timely manner (*Figure 9—figure supplement 1*). As both the Sgo1-100 and Sgo1-700 proteins themselves fail to be maintained at the centromere (*Figure 1B*), these observations are difficult to interpret. As a more direct test of the ability of the mutant proteins to recruit condensin, we fused Sgo1-100, Sgo1-700, and Sgo1-3A to tetR and artificially tethered them to *tetO* arrays located adjacent to *CEN4* in cells that otherwise lacked *SGO1* (*Figure 9D–F*). Tethering of Sgo1-tetR-GFP, Sgo1-100-tetR-GFP, or Sgo1-3A-tetR-GFP, all significantly increased Brn1-6HA levels at *CEN4* compared to a no tag control (*Figure 9E*). In contrast, and consistent with our finding that Brn1 fails to co-immunoprecipitate with Sgo1-700 (*Figure 9B*), tethering of Sgo1-700-tetR-GFP did not significantly enrich Brn1-6HA at the same site, though it was produced to similar levels as the Sgo1-100-tetR-GFP protein (*Figure 9D,E*). Despite the ability of Sgo1-100-tetR-GFP and Sgo1-3A-tetR-GFP fusion proteins to recruit Brn1 to the tethering site at *CEN4*, only the wild-type Sgo1-tetR-GFP fusion protein was able to partially rescue the segregation of chromosome IV after release from a nocodazole arrest (*Figure 9—figure supplement 2*). This is anticipated as none of the mutant proteins are expected to enable proper Ipl1 association with the centromere (*Figure 3E*). We conclude that Sgo1-100 and Sgo1-3A are able to associate with, and recruit, condensin, whereas Sgo1-700 loses this interaction.

This finding allowed us to test a prediction: if Sgo1-dependent deposition of condensin at the pericentromere is critical for biasing sister kinetochores towards biorientation, then Sgo1-700, which fails to bind to or recruit condensin to the pericentromere (*Figure 9B–E*) should lack the sister kinetochore bias. In contrast, Sgo1-3A, which recruits condensin to the pericentromere (*Figure 9B,C*), and Sgo1-100, which retains at least a partial ability to deposit condensin at the pericentromere (*Figure 9F*, *Figure 9—figure*

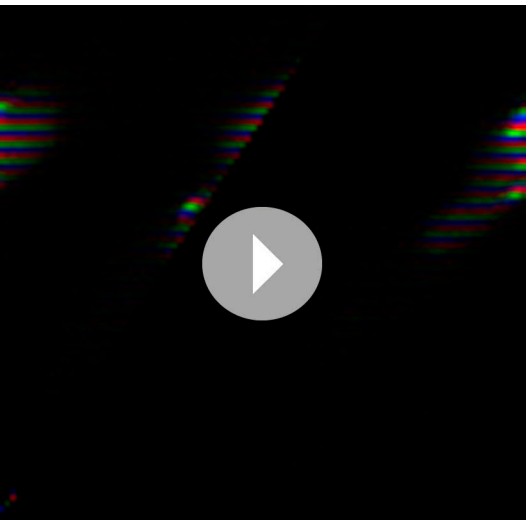

**Video 1**. Example video of a wild-type cell in the error correction assay. The video corresponds to the image gallery in *Figure 5C* (upper panel).

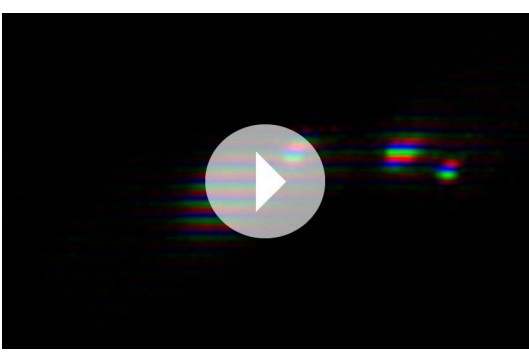

**Video 2**. Example video of an ipl1-as cell in the error correction assay. The video corresponds to the image gallery in *Figure 5C* (lower panel).

*supplement 1*), should enable some degree of sister kinetochore bias. In accordance with these predictions, our live cell assay revealed that only the *sgo1*-700 mutant was significantly defective in the sister kinetochore bias (*Figure 9G*). We conclude that condensin recruitment by Sgo1 biases sister kinetochores towards biorientation.

## Discussion

Our findings have demonstrated that Sgo1 plays a central role in promoting biorientation, through at least two separate mechanisms. Sgo1 enables error correction by retention of Ipl1 (aurora B) at the centromere. Sgo1 separately recruits condensin to the pericentromere, which both imposes a bias on sister kinetochores to biorient and enables effective error correction.

Our findings provide the first molecular explanation for the bias to sister kinetochore biorientation. Interestingly, *Indjeian and Murray (2007)* demonstrated the existence of a bias to sister kinetochore biorientation in the *sgo1-100* mutant, which is defective in the error correction process (*Indjeian et al., 2005*). Consistent with this study, we confirmed that the *sgo1-100* mutant retains this bias to sister kinetochore biorientation (*Figure 9G*). Through analysis of both the *sgo1-700* and *sgo1Δ* mutants we have, nevertheless uncovered a role for Sgo1 in biasing sister kinetochores towards biorientation. Moreover, our data provide a molecular explanation for the loss of sister kinetochore bias in *sgo1-700* and *sgo1Δ* cells: that is the inability to associate with and recruit condensin to centromeres. In contrast, *sgo1-100* and *sgo1-3A* cells, which can enable condensin recruitment to centromeres are proficient in the bias to sister kinetochore biorientation.

Condensin is a cohesin-related complex that can form a ring-like structure and has been proposed to organize chromosomes by bringing distant chromosomal sequences together (*Cuylen and Haering, 2011*; *Cuylen et al., 2011*; *Hirano, 2012*). The mechanism by which condensin is loaded onto and subsequently maintained on chromosomes is largely unknown. Here, we have identified shugoshin as an important determinant of condensin association with the pericentromere. Indeed, the pericentromeres and rDNA appear to be the predominant, if not the only sites of condensin association with chromosomes in mitosis. Our findings have also suggested a hierarchy of assembly at the pericentromere. In contrast to the idea that the Scc2/4 cohesin-loader complex loads condensin directly (*D'Ambrosio et al., 2006*), we propose that Scc2/4 indirectly affects condensin localization through loading cohesin, which we show is required for the pericentromeric association of Sgo1. In support of this idea, only cohesin and not condensin subunits were identified in Scc2/4 immunoprecipates (*Fernius et al., 2013*). Furthermore, since Shugoshin relies on cohesin for its association with the pericentromere (*Kiburz et al., 2005*; *Yamagishi et al., 2008*; *Figure 5*), the importance of cohesin in defining kinetochore geometry (*Sakuno et al., 2009*) could be to enable proper condensin association with chromosomes by recruiting shugoshin to the pericentromere. How might condensin shape a favorable pericentromeric geometry? While the molecular function of condensin is not well understood, condensin is known to alter the structural properties of centromeres that affect their dynamic behavior (*Ribeiro et al., 2009*; *Stephens et al., 2011*). We propose that condensin organizes the pericentromeric chromatin to provide a structural integrity to the pericentromere that enables it to adopt a 'back-to-back' geometry that orients sister kinetochores in opposite directions, thereby favoring their biorientation (*Figure 10*). Although our data indicate that condensin is important to increase the efficiency of biorientation, we do not believe that it is essential for this process in budding yeast, provided that error correction machinery is intact. However, budding yeast centromeres are relatively simple and only a single microtubule contacts each kinetochore (*Winey et al., 1995*) increasing the probability that correct attachments can be made by chance. This is not the case for organisms with more complex centromeres where it is likely that kinetochore geometry plays a major role in ensuring sister kinetochore biorientation is achieved.

In addition to influencing pericentromeric structure through condensin, shugoshins also confer distinct properties to pericentromeric cohesin through PP2A, both in meiosis and mammalian mitosis,

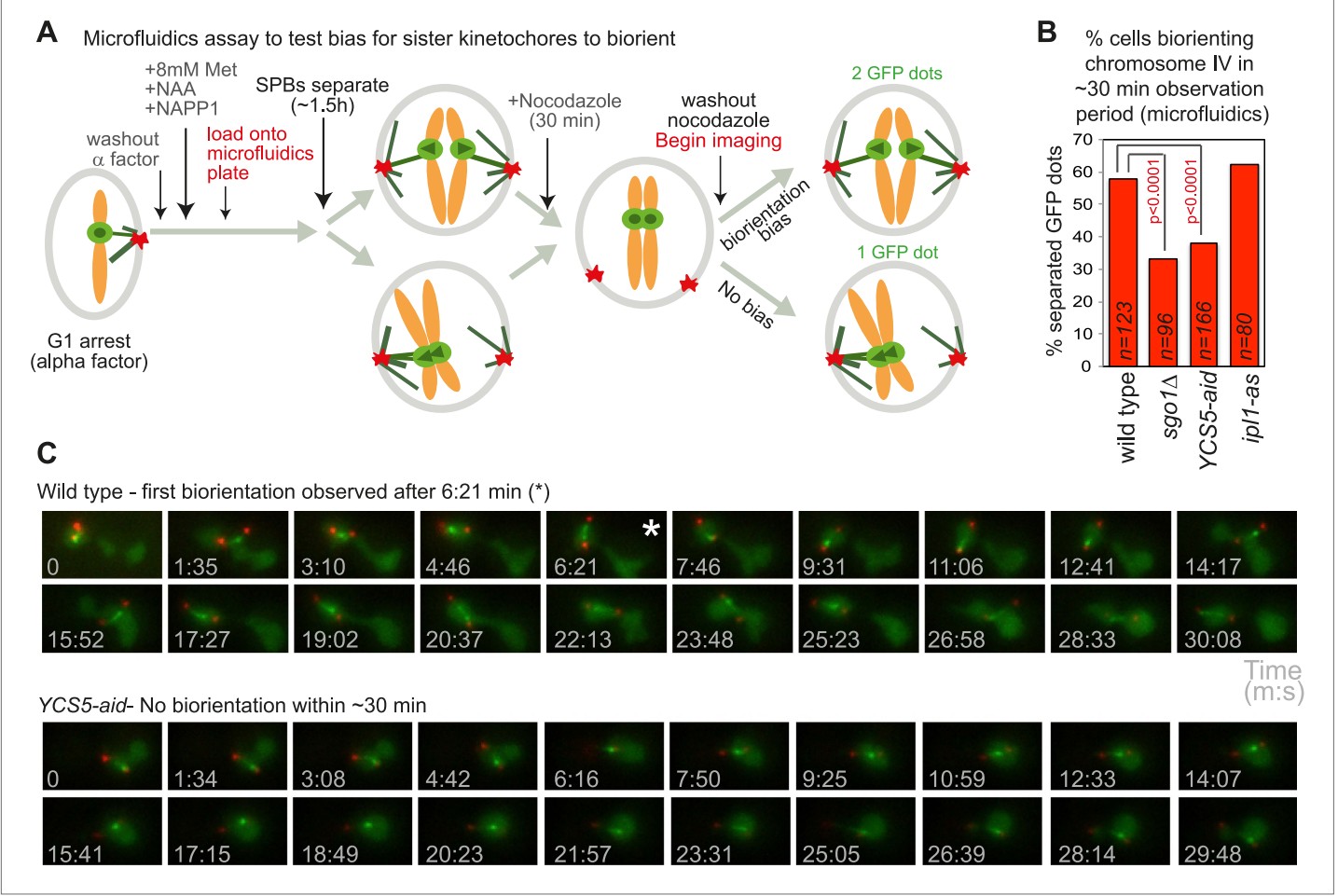

**Figure 8**. Condensin biases chromosomes to biorient. (**A**–**C**) Condensin and Sgo1, but not Ipl1, are required to bias sister kinetochores towards biorientation. (**A**) Scheme of the microfluidics assay to test sister kinetochore bias. Wild-type, *sgo1Δ*, *YCS5-aid* and *ipl1-as* cells as in **Figure 5B** were released from a G1 arrest into NAA and NAPP1 and arrested in metaphase by *CDC20* depletion. SPBs were allowed to separate for 1 hr 30 min before cells were treated with nocodazole for an additional 30 min. After 2 hr total, nocodazole was washed out and frames were grabbed at approximately 94 s intervals for a total of 21 frames. (**B**) The percentage of cells that separated *CEN4-GFP* foci at least once during the observation period is shown for the indicated strains. p values indicate significance (chi-square test). (**C**) Representative images of wild-type and *YCS5-aid* cells are shown. Time of image acquisition (s) is shown. The asterisk indicates the first time GFP foci are separated.

as well as influence kinetochore–microtubule interactions through aurora B. Shugoshins are therefore emerging as functional hubs that define the pericentromere, allowing it to perform specialized functions that are key for the fidelity of chromosome segregation.

## Materials and methods

### Yeast strains

Strains used in this work are listed in **Supplementary file 2A**. All yeast strains were derivatives of W303 except the protease-deficient strain, JB811, used for TAP pulldowns. The *sgo1-100* and *sgo1-700* alleles were described in **Indjeian et al. (2005)**, the *sgo1-3A* allele was described in **Xu et al. (2009)** and *sgo1Δ* was described in **Clift et al. (2009)**. A PCR-based approach was used to tag Sgo1 with SZZ(TAP); Ipl1, Brn1 and Ycs4 with 6HA; and Rts1 with 3Pk or 9Myc (**Longtine et al., 1998**; **Knop et al., 1999**). Auxin-inducible degron versions of Sgo1 and Ycs5 were constructed as described by **Nishimura et al. (2009)**. To generate a strain carrying Sgo1-TetR-GFP, *SGO1* was cloned upstream of *tetR-GFP* in p128(TetR-GFP) (**Michaelis et al., 1997**) to generate plasmid AMp769 which was integrated at the *LEU2* locus after *EcoRV* digestion. TetR-GFP fusions to Sgo1-100,

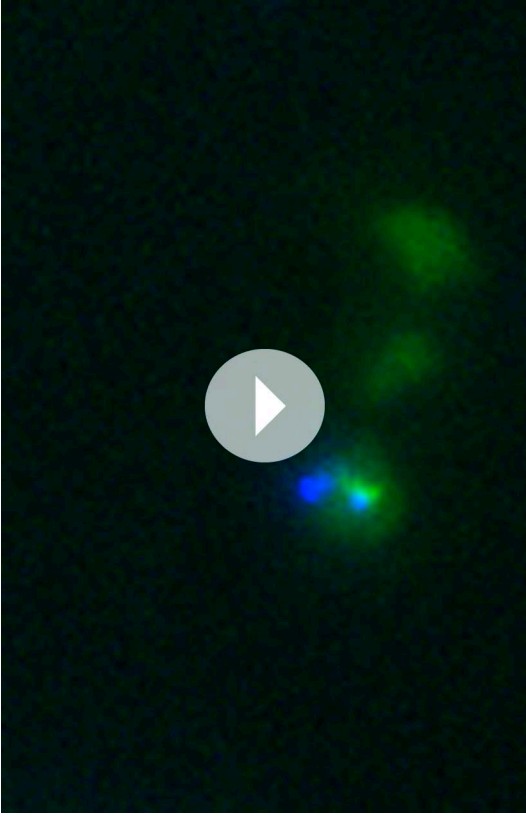

**Video 3**. Example video of a wild type cell in the assay to test for a bias towards sister kinetochore biorientation. The video corresponds to the image gallery in **Figure 6C** (upper panel).

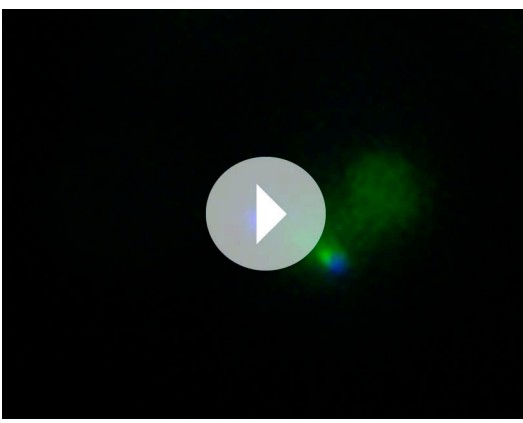

**Video 4**. Example video of a YCS5-aid cell in the assay to test for a bias towards sister kinetochore biorientation. The video corresponds to the image gallery in **Figure 6C** (lower panel).

Sgo1-700 or Sgo1-3A were generated by PCR amplification of these alleles from the genomic locus and replacement of *SGO1* in AMp769 (*sgo1-700, sgo1-3A*) or by site-directed mutagenesis using the Quikchange kit (Agilent Technologies, Santa Clara, CA). *Ipl1-as5* was described in *Pinsky et al. (2006)*. *pMET-CDC20* was described in *Fernius and Marston (2009)*. The *CEN4-GFP* label (*CEN4-tetOs::URA3 leu2::TetR-GFP::LEU2*) was described in *He et al. (2000)*; *Tanaka (2010)* and *SPC42-tdTomato* was described in *Fernius and Hardwick (2007)*.

## Growth conditions

Nocodazole was used at 15 μg/ml and re-added to 7.5 μg/ml every 1 hr. NAA (synthetic auxin) was used at 500 μM and readded to 250 μM every 45 min. Methionine was used at 8 mM and re-added to 4 mM every 45 min. NAPP1 (to inhibit *ipl1-as*) was used at 50 μm and doxycycline was used at 5 μg/ml.

## Chromatin immunoprecipitation

ChIP was performed as described using anti-HA 12CA5, anti-Myc 9E10 or anti-Pk(V5) antibody (*Fernius et al., 2013*). Primers used for qPCR analysis are given in *Supplementary file 2B*. qPCR was performed in a 20 μl Express SYBR GreenER (Life Technologies, Carlsbad, CA) reaction using a Lightcycler machine (Roche, Switzerland). To calculate ChIP enrichment/input, $\Delta CT$ was calculated according to: $\Delta CT = (CT_{(ChIP)} - [CT_{(Input)} - logE$ (Input dilution factor)]) where E represents the specific primer efficiency value. Enrichment/ input value was obtained from the following formula: $E^{-\Delta CT}$. qPCR was performed in triplicate, typically for each of three or more independent experimental repeats. Error bars represent standard error. For ChIP-Seq, purified chromatin was recovered using a PCR purification kit (Qiagen, Netherlands) followed by drying in a speedvac. Samples were sequenced on a HiSeq2000 instrument (Illumina, San Diego, CA) by the EMBL Core Genomics Facility (Heidelberg, Germany). The summary of data obtained is given in *Supplementary file 2C*.

## Analysis of ChIP-Seq data

Scripts, data files and workflows used to create the ChIP-Seq data can be found on the github repository: https://github.com/AlastairKerr/Marston2013. Single reads were mapped using BWA (Version: 0.6.1-r104) (*Li and Durbin, 2010*) to the sacCer3 reference genome and were processed with samtools (*Li et al., 2009*) to remove duplicated reads for parallel analysis. The region of chrXII containing rDNA (451400 bp to 490600 bp) was removed and studied separately using bedtools (version v2.16.2) (*Quinlan and Hall, 2010*). As this region is highly repetitive and contained a high density of reads,

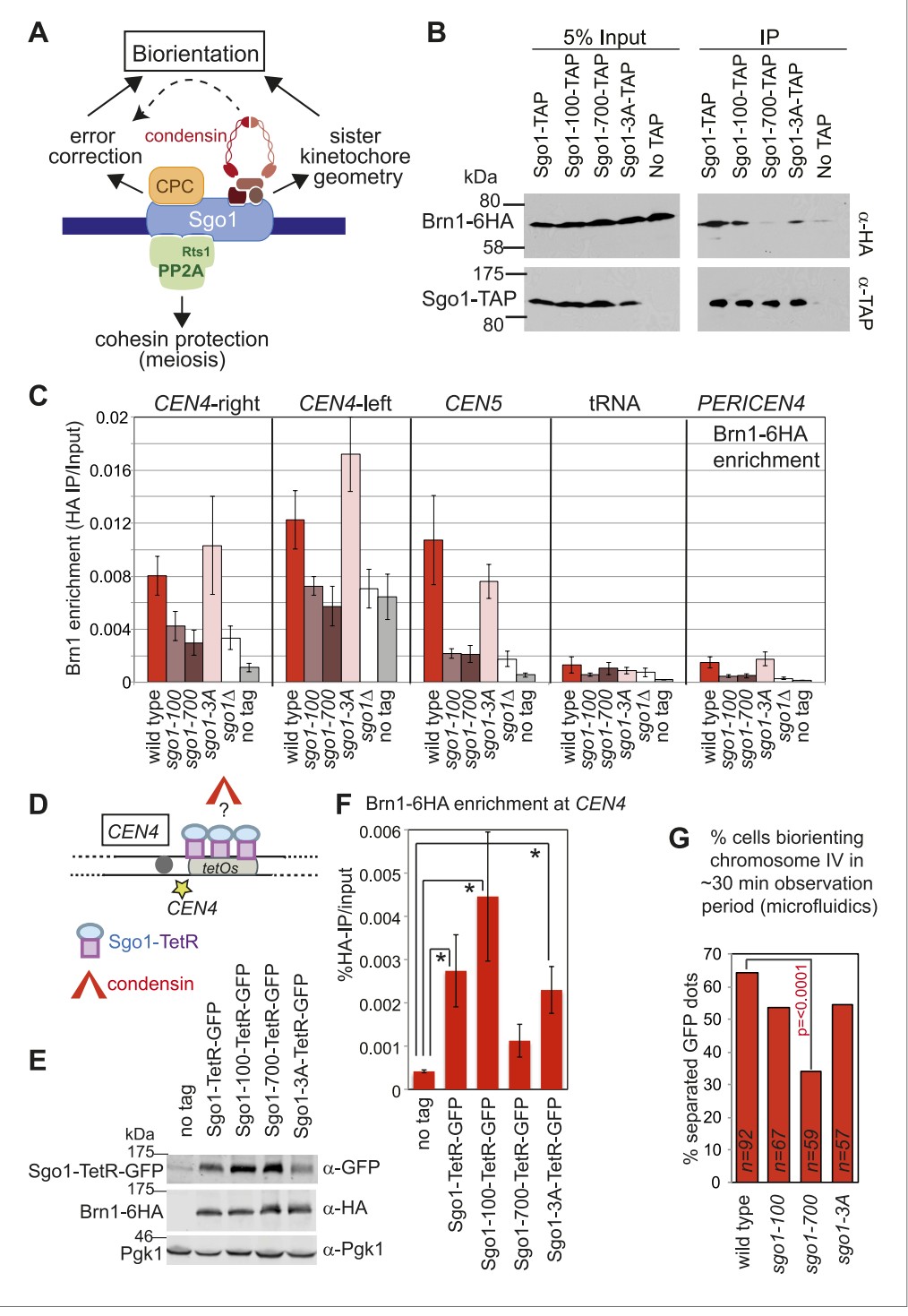

**Figure 9**. Shugoshin enables the bias towards sister kinetochore biorientation by condensin recruitment. (**A**) Schematic diagram showing Sgo1-associated complexes and their functions at the pericentromere. (**B**) Sgo1-100 and Sgo1-3A, but not Sgo1-700, retain association with Brn1. Cells carrying *BRN1-6HA* and *SGO1-SZZ(TAP)* (AM9266), *SGO1-100-SZZ(TAP)* (AM9149), *SGO1-700-SZZ(TAP)* (AM9264), *SGO1-3A-SZZ(TAP)* (AM9262) or no TAP (AM5708) were arrested in nocodazole for 2 hr before treating with the cross-linker DSP. Prepared extracts were incubated with IgG-coupled beads and immunoprecipitates analyzed by immunoblotting with the indicated antibodies. (**C**) Brn1 is maintained at the centromere in metaphase-arrested *sgo1-3A*, but not *sgo1-100* or *sgo1-700* cells. Wild-type (AM5708), *sgo1Δ* (AM8834), *sgo1-100* (AM9442), *sgo1-700* (AM9291) and *sgo1-3A* (AM9276) cells

*Figure 9. Continued on next page*

*Figure 9. Continued*

carrying *BRN1-6HA* as well as a no tag control (AM1176) were arrested in nocodazole for 2 hr before harvesting for anti-HA ChIP. The levels of Brn1-6HA were measured at the indicated sites by qPCR. (**D**–**F**) Tethered Sgo1, Sgo1-100 or Sgo1-3A, but not Sgo1-700 can enrich Brn1-6HA at *CEN4* in otherwise *sgo1Δ* cells. *SGO1-tetR-GFP* (AM14012), *sgo1-100-tetR-GFP* (AM13902), *sgo1-700-tetR-GFP* (AM13907), and *sgo1-3A-tetR-GFP* (AM13904) were introduced into cells carrying *tetOs* integrated at *CEN4*, producing Brn1-6HA and with *SGO1* deleted from its endogenous locus. A strain carrying just *tetOs* integrated at *CEN4* but otherwise wild type was used as a no tag control (AM11060). All strains were arrested in mitosis by treatment with nocodazole for 3 hr before harvesting for ChIP and immunoblotting. (**D**) Schematic diagram of the tethering locus. (**E**) Levels of Brn1 recruited adjacent to the tethering site (*CEN4*) when the indicated proteins are fused to TetR-GFP, as measured by ChIP-qPCR. The mean of four independent repeats is shown except for *sgo1-3A-tetR-GFP* where six repeats are included. Error bars are standard error, significance was calculated using the student *t* test (*p<0.05). (**F**) Total cellular levels of the Sgo1-tetR-GFP fusion proteins, Brn1-6HA and Pgk1 (loading control) were analyzed by immunoblot using the indicated antibodies. (**G**) The bias to sister kinetochore biorientation is absent in *sgo1-700* cells. Wild-type (AM4643), *sgo1-100* (AM8924), *sgo1-700* (AM8925) and *sgo1-3A* (AM8923) cells were released from G1 and treated with nocodazole after SPB separation as in *Figure 7A*. The percentage of cells that separated *CEN4-GFP* foci at least once during the observation period is shown for the indicated strains. p values indicate significance (chi-square test).

The following figure supplements are available for figure 9:

**Figure supplement 1**. The Sgo1-100 protein can recruit condensin to kinetochores.

**Figure supplement 2**. Sgo1-tetR-GFP, but not Sgo1-100-tetR-GFP, Sgo1-700-tetR-GFP or Sgo1-3A-tetR-GFP tethered to *CEN4* can partially rescue the mis-segregation of chromosome IV after nocodazole washout in otherwise *sgo1Δ* cells.

---

duplicate reads were not removed for this analysis as it would not be possible to differentiate duplicate reads from independent fragments. All data shown are normalized to the number of mapped reads per million total mapped reads [RPM]. Total mapped reads were calculated after any processing was done for rDNA or duplicate read removal.

During the amplification step prior to sequencing, multiple identical reads will be generated. Due to the low yield of total DNA precipitating with Brn1-6HA in *sgo1Δ* cells, we were concerned that the small number of precipitating sequences would be biased for amplification. To avoid this problem, we removed PCR replicates from our analysis. However, since this would also remove bona fide identical sequences from our dataset, we simultaneously analyzed the data with these reads present and compared the two data sets where all 16 centromeres were considered together. To do this, 100 bp windows were examined that extended 25 kb in each direction of every centromere, and a local maximum of the number of mapped reads was taken. The comparison of this data with and without PCR replicates included can be seen in *Figure 3H* (unfiltered data) and *Figure 4—figure supplement 2* (PCR replicates removed). Both approaches led to very similar conclusions, therefore for all other data presented we used the unfiltered data.

## TAP pulldowns and mass spectrometry

To grow cells for purification of Sgo1 from cycling cells, 4 L of YPDA culture were inoculated to $OD_{600}$ = 0.2 and grown at room temperature for 6 hr. Cells were harvested by centrifugation and the cell pellet was resuspended in 2 µM of dithiobis(succinimidylpropionate) (DSP; Proteochem, Loves Park, IL) crosslinker for 30 min at room temperature before cell pellets were drop-frozen in liquid nitrogen as described by *Fernius et al. (2013)*. To purify Sgo1 from cells arrested in mitosis using the cold-sensitive *tub2-401* allele, 4 L of media was inoculated to $OD_{600}$ = 0.02, grown at 30°C for 8 hr with shaking and then the temperature was reduced to 18°C for 7 hr to induce the arrest before holding the cells at 4°C for up to 5 hr before harvesting, crosslinking, and freezing as described above. Pulldowns and mass spectrometry were performed as described in *Fernius et al. (2013)*.

## Co-immunoprecipitation, western blotting, immunofluorescence, and FACS

For co-immunoprecipitation of TAP-tagged Sgo1, 200 ml of nocodazole-treated culture was harvested and either firstly cross-linked using DSP as described in *Fernius et al. (2013)*, or directly drop-frozen in liquid nitrogen. Frozen pellets were ground in a pestle and mortar for 5 min. Ground lysates, prepared

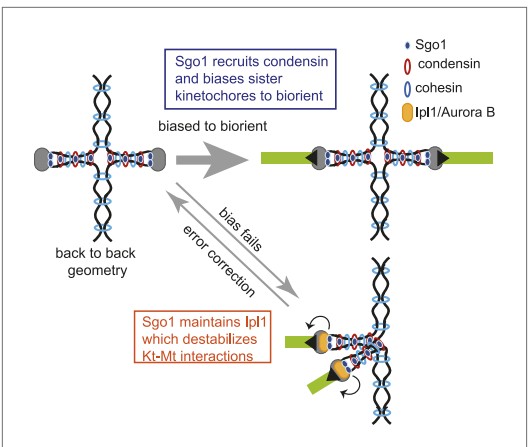

**Figure 10**. Model for dual role of Sgo1 in biorientation is shown. Shugoshin ensures sister kinetochore biorientation through two mechanisms. First, early in the cell cycle, shugoshin mediates the enrichment of condensin within the pericentromere. We propose that condensin enables the pericentromere to adopt a geometry that favors the capture of sister kinetochores by microtubules from opposite poles, thereby biasing them to biorient. Second, shugoshin maintains aurora B at the pericentromere for those kinetochores that fail to biorient and come under tension. Aurora B destabillizes these tension-less attachments, thereby providing a further chance to for sister kinetochores to make the appropriate, tension-generating attachments. We suggest that condensin facilitates this 'error correction' process by conferring a rigid structure to the pericentromere upon which aurora B can act.

as in *Fernius et al. (2013)*, were incubated with 1 mg of IgG-coupled dynabeads for 1.5 hr at 4°C. For the DNAase treated samples in *Figure 4—figure supplement 1*, benzonase (25U) was added to the extracts before incubating at room temperature for 30 min before proceeding with immunoprecipitation. Protein complexes were eluted from the beads by addition of 30 µl of sample buffer and loaded onto polyacrylamide gels.

Western immunoblot was performed as described in *Clift et al. (2009)* and visualized by detection of chemiluminesence on autoradiograms except for the data shown in *Figure 9E* where proteins were visualized using a fluorophore-conjugated antibody and the Odyssey system (Li-Cor, Lincoln, NE). Mouse anti-aid (CosmoBio, Japan), anti-HA11, and anti-Myc antibodies were all used at a dilution of 1:1000. Mouse anti-Pgk1 antibodies were used at a dilution of 1:5000. Rabbit anti-Sgo1 (a kind gift of Adam Rudner, Ottawa Institute of Systems Biology, Canada), anti-GFP (a kind gift of Eric Schirmer, University of Edinburgh) and anti-Pgk1 antibodies were used at a dilution of 1:5000. Rabbit anti-Kar2 antibodies (laboratory stock) were used at a dilution of 1:5000.

Indirect immunofluorescence and FACS were performed as described in *Clift et al. (2009)* and *Fernius and Marston (2009)*, respectively.

## Biorientation assay

Cells carrying *pMET-CDC20*, *CEN4-GFP* and *SPC42-tdTomato* were arrested in G1 using alpha factor (4 µg/ml) in minimal medium lacking methionine (SD/-met) at room temperature for 3 hr. Alpha factor was washed out and cells were released from the G1 arrest into rich medium containing methionine and nocodazole (YPDA + Met + NOC) plus the appropriate drugs (NAPP1, NAA). Methionine represses *pMET-CDC20* expression, resulting in a metaphase arrest and nocodazole depolymerizes microtubules. After 3 hr, a sample was extracted (t = 0) and nocodazole was washed out by filtering in the presence of methionine and cultures were released into YPDA + Met to allow spindles to reform while maintaining the metaphase arrest. Samples were taken at 20 min intervals. Samples were fixed in 3.7% formaldehyde for 10 min, before washing in PBS and resuspending in DAPI for microscopy. Samples with two separated SPC42-tdTomato foci were scored as containing one or two *CEN4-GFP* foci. Typically 200, and at least 100, cells were scored for each timepoint.

## Microfluidics and image analysis

For live cell imaging, cells were loaded onto the Onix Microfluidic Perfusion system (CellAsic, Hayward, CA) and visualized using a Deltavision Elite (Applied Precision, Issaquah, WA) coupled to a Cascade 2 EMCCD camera with temperature control to 30°C. Frames were grabbed at the indicated intervals and images were processed in ImagePro software (Media Cybernetics, Rockville, MD).

For image analysis, a custom-written macro was developed in ImagePro software (Media Cybernetics), details of which are available upon request. In brief, yeast are selected from a reference DIC image taken at each time point. The yeast are automatically thresholded and the centre pixel from each red and green spot is then calculated. The distances between the spots can then be calculated and the measurements output to a text file for analysis using microsoft excel. Measurements of inter-centromere distance and distance from *CEN4*-GFP dot to nearest Spc42-tdTomato were obtained using this automated system.

## Error correction assay using microfludics

To test the efficiency of error correction, cells carrying CEN4 (CEN4-GFP) and SPB (SPC42-tdTomato) markers together with CDC20 under control of a methionine-repressible promoter (pMET-CDC20) were arrested in G1 using alpha factor and then loaded onto the microfluidic chamber. Cells were released in the chamber into medium containing nocodazole (to depolymerize microtubules) and methionine (to deplete CDC20 and induce a metaphase arrest) for 30 min. The addition of nocodazole to G1 cells prevents SPB separation because this requires microtubules. Therefore, under these conditions, microtubules do not separate which leads to a high rate of monoorientation once microtubules are allowed to reform. This leads to a strong reliance on the error correction machinery to establish biorientation (Indjeian and Murray, 2007). After 30 min, nocodazole was washed out, all the time maintaining the CDC20 arrest by inclusion of methionine in the media. After 90% of cells had 2 SPBs (typically 1.5 hr), we began imaging at ~74s intervals for a total of 21 frames.

For determination of the overall ability of strains to biorient in Figure 2D all cells in each frame where two SPB foci were detected by the software were scored for the presence of either one or two CEN4-GFP foci. The total percentage of cells with two SPB foci that contained two GFP foci was calculated for all frames for each strain. The number of cell images analyzed was 1116 (wild type), 697 (sgo1Δ), 1251 (YCS5-aid) and 968 (ipl1-as). To m

As a measure of the efficacy of the error correction machinery, we determined the ability of cells to switch between one and two visible CEN4-GFP foci. Cells in which two SPB foci were detected for at least four consecutive frames were scored for the number of times that the number of CEN4-GFP foci changed from one to two or vice-versa for the frames in which two SPBs were consecutively visible. The 'switching rate' was calculated by dividing the number of times a cell alternated between one and two CEN4-GFP foci by the total time in which two SPB foci were consecutively visible. The average switching rate was determined for all cells in which two SPB foci were detected for at least four consecutive frames. In Figure 2F, we analyzed 92 wild-type cells (971 frames), 55 sgo1Δ cells (551 frames), 102 YCS5-aid cells (1046 frames) and 86 ipl1-as cells (748 frames).

## Sister kinetochore bias assay using microfludics

To test the bias on sister kinetochores to biorient, cells carrying CEN4 (CEN4-GFP) and SPB (SPC42-tdTomato) markers together with CDC20 under control of a methionine-repressible promoter (pMET-CDC20) were arrested in G1 using alpha factor and then released into medium containing methionine to deplete CDC20 and induce a metaphase arrest. Cells were loaded onto the microfluidics chamber and after approximately 1.5 hr, when around 90% of cells had 2 SPBs, cells were treated with YPDA containing nocodazole for 30 min. After 30 min, nocodazole was washed out after which cells were immediately imaged every ~94 s for a total of 21 frames. We scored the percentage of cells in which separated GFP dots were observed at least once during the observation period.

## Acknowledgements

We are grateful to Adam Rudner for the kind gift of anti-Sgo1 antibody, to Eric Schirmer for the anti-GFP antibody and Andrew Murray and Kim Nasmyth for yeast strains. We thank Bianka Baying at Genecore, EMBL for library preparation and sequencing. We appreciate help from Claudia Schaffner with FACS. We are grateful to Robin Allshire, Kevin Hardwick, and Vasso Makrantoni for helpful comments on the manuscript.

## Additional information

### Funding

| Funder | Grant reference number | Author |
|---|---|---|
| Wellcome Trust | [090903], [084229], [077707], [092076], [091020] | Kitty F Verzijlbergen, Olga O Nerusheva, David Kelly, Alastair Kerr, Dean Clift, Flavia de Lima Alves, Juri Rappsilber, Adele L Marston |

| Funder | Grant reference number | Author |
| --- | --- | --- |
| EMBO | | Kitty F Verzijlbergen, Adele L Marston |
| Scottish Universities Life Sciences Alliance | | Adele L Marston |
| Darwin Trust of Edinburgh | | Olga O Nerusheva |

The funders had no role in study design, data collection and interpretation, or the decision to submit the work for publication.

## Author contributions

KFV, ALM, Conception and design, Acquisition of data, Analysis and interpretation of data, Drafting or revising the article; OON, Acquisition of data, Analysis and interpretation of data; DK, Wrote a macro for analysis of microscopy data, Assisted with acquisition and analysis of live cell imaging data; AK, Bioinformatics associated with ChIP-Seq data, Analysis and interpretation of data; DC, FLA, JR, Acquisition of data

## Additional files

### Supplementary files

• Supplementary file 1. Complete list of peptides identified in the experiments shown in *Figure 4A,B*.

• Supplementary file 2. (**A**) Yeast strains used in this study. (**B**) qPCR primers used in this study. (**C**) Genome summary table for Brn1-6HA ChIP-seq.

### Major dataset

The following dataset was generated:

| Author(s) | Year | Dataset title | Dataset ID and/or URL | Database, license, and accessibility information |
| --- | --- | --- | --- | --- |
| Verzijlbergen K, Kerr A, Marston A | 2013 | Shugoshin biases chromosomes for biorientation through condensin recruitment to the pericentromere | GSE53856; http://www.ncbi.nlm.nih.gov/geo/query/acc.cgi?acc=GSE53856 | Publicly available at the Gene Expression Omnibus (http://www.ncbi.nlm.nih.gov/geo/). |

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
