## [Decision Letter]

Thank you for sending your work entitled “Shugoshin biases chromosomes for biorientation through condensin recruitment to the pericentromere” for consideration at *eLife*. Your article has been favorably evaluated by a Senior editor and 3 reviewers, one of whom is a member of our Board of Reviewing Editors.

The Reviewing editor and the other reviewers discussed their comments before we reached this decision, and the Reviewing editor has assembled the following comments to help you prepare a revised submission.

In this study the authors have investigated how sister kinetochores in budding yeast are biased towards biorientation. The authors have found that the Shuoshin protein (Sgo1) is important to promote biorientation through two mechanisms: through binding to the chromosome passenger complex, and through a newly identified association with the condensin complex. The authors show that Sgo1 is able to recruit condensin to centromeres or to ectopic sites. The authors identify a mutation in Sgo1 that perturbs binding to condensin and that this association is important for subsequent biorientation. The authors conclude that condensin is required to form a rigid back-to-back structure such that sister kinetochores are biased towards microtubule capture from opposite spindle poles.

Overall this is an interesting study that reveals new insights into how budding yeast chromosomes biorent, and uncover a new role for condensin in centromere and kinetochore behaviour.

There are, however, a number of substantive concerns that preclude publication in its present form. In particular, the conclusion that condensin recruitment alters kinetochore geometry to promote orientation towards opposite poles requires further supporting evidence. Moreover, the authors themselves do not appear to be certain whether there is a direct link between Sgo1 and condensin recruitment. They argue that Sgo1 is sufficient for condensin targeting to the pericentromere by over-expressing Sgo1 and measuring Brn1 accumulation. However, they previously state that Sgo1's effect on condensin is indirect, mediated through Scc2/4 (cohesin). There are also concerns regarding the validity of the approach the authors use to measure bi-orientation and the presentation of the ChIP analyses. The latter prevents a critical evaluation of the data quality and missing information will need to be added.

Concerns to be addressed:

1) The evidence for a direct interaction between Sgo1 and condensin is limited. Only a small number (especially considering the large size of all five condensin subunits) of condensin peptides can be identified in the mass-spec analysis of Sgo1 purifications. In fact, the highest scoring condensin subunit (by numbers of found peptides; Ycg1) ranks number 137 in the list of proteins identified (see supplemental Excel file). Moreover, only a minute fraction of condensin (less than 0.01%? See Figure 3) can be detected by Western blotting in Sgo1 immunoprecipitations. Since both Sgo1 and condensin are enriched at centromeres, it is very likely that a small amount of condensin co-precipitates with Sgo1 because both proteins are bound to the same stretch of chromatin. In addition, the authors add amine-reactive crosslinkers to the cell extracts in both experiments (a fact they should mention at least in the figure legends and not only in the methods section), which will result in the formation of protein-chromatin crosslinks. The authors will need to repeat the pull-down experiments after DNase digestion and in the absence of crosslinker in order to rule out these concerns.

A second argument for a role of Sgo1 in condensin loading is that centromeric condensin levels increase upon Sgo1 overexpression. The increase in condensin binding could, however, result from the delay in anaphase onset caused by Sgo1 overexpression. Would condensin levels remain unchanged if anaphase onset were delayed by other means?

Ectopic recruitment of condensin by Sgo1 fused to TetR seems rather inefficient, with a two-fold enrichment (and considerable variation?). The authors should at least test whether ectopic recruitment can be measured at both sides of the TetO repeats. Yet a much better test whether condensin recruitment by Sgo1 is significant would be to employ the TetO-TetR approach to tether to centromeres an Sgo1 mutant that can otherwise not bind (e.g., Sgo1-100 or Sgo1-700) and assay whether this restores condensin levels and chromosome bi-orientation.

An explanation for the altered binding profiles of condensin measured by ChIP-seq might be changes in cohesin accumulation at centromeres in the Sgo1 mutants. The authors need to check by ChIP-qPCR (and ideally also ChIP-seq) whether cohesin levels at centromeres are affected in the Sgo1 mutant.

2) The authors monitor the splitting of a sister centromere pair as a measure for sister chromatid bi-orientation (Figures 1 and 5, Figure 3—figure supplement 2). The reduction of sister centromere splitting observed in the Sgo1 and condensin mutants could, however, equally well result from an increase in centromeric sister chromatid cohesion. An increase in centromeric cohesion would prevent splitting of bi-oriented sister chromatids.

One possibility to distinguish whether chromatid pairs have attached to spindle microtubules from only one spindle pole or whether chromatids have bi-oriented but their centromeres cannot separate would be to score centromere-spindle pole body (SPB) distances in the live cell microscopy assay (Figure 5). If centromeres were bi-oriented but merely failed to split, they should be positioned at equal distances to both SPBs. If sister centromeres failed to bi-orient, they should frequently be found close to one SPB. Alternatively, the authors could release cells from the Cdc20 arrest and allow them to enter anaphase, then score whether sister centromeres segregate to one pole or to opposite poles (see Indjeian and Murray, CurBiol 2007).

The data in Figures 6 and 7 show a mild, real defect in biorientation but don't specifically address the of kineteochore geometry. The authors should either perform further experiments to support their hypothesis or confine their “geometric bias” to speculation in the Discussion.

3) Most of the graphs that represent the results from ChIP-qPCR experiments (Figures 1, 2, 5 and 7, Figure 5—figure supplement 3) lack information that is essential to evaluate the data. What do the column plots represent? Do they show mean values, and if yes, from how many PCR runs and how many independent immunoprecipitation experiments? What do the error bars represent? The legend for Figure 4 is missing completely.

Taking into account the possible variability between individual ChIP-qPCR experiments, careful statistical analysis will be required to evaluate the significance of the observed differences. For example, the values measured at CEN4 for the same wild-type strain (AM3513) in two independent Ipl1 HA-ChIP experiments vary 4-fold (compare Figure 2), calling into question whether the 2-fold reduction in ChIP signals in the Sgo1 mutants (Figure 2) is significant. Similarly, the ChIP signal at CEN4 measured for the Sgo1-100 mutant in G1 phase is more than 2-fold higher than the signal for wildtype Sgo1 (Figure 1). Is the 4-fold increase in Sgo1 ChIP signals during mitosis (e.g., 2-fold when compared to the Sgo1-100 mutant) then still significant?

---

## [Author Response]

*1) The evidence for a direct interaction between Sgo1 and condensin is limited. Only a small number (especially considering the large size of all five condensin subunits) of condensin peptides can be identified in the mass-spec analysis of Sgo1 purifications. In fact, the highest scoring condensin subunit (by numbers of found peptides; Ycg1) ranks number 137 in the list of proteins identified (see supplemental Excel file)*.

The Excel file submitted with the original manuscript was unsorted and did not take the contaminant proteins that also co-purify with the “no tag” control into account. When contaminants are taken into account, the condensin subunits rank number 8 and 26 in the list of proteins identified, in cycling and arrested cells, respectively. We have revised the file with the list of proteins sorted to exclude contaminant proteins.

*Moreover, only a minute fraction of condensin (less than 0.01%? See*
Figure 3*) can be detected by Western blotting in Sgo1 immunoprecipitations. Since both Sgo1 and condensin are enriched at centromeres, it is very likely that a small amount of condensin co-precipitates with Sgo1 because both proteins are bound to the same stretch of chromatin. In addition, the authors add amine-reactive crosslinkers to the cell extracts in both experiments (a fact they should mention at least in the figure legends and not only in the methods section), which will result in the formation of protein-chromatin crosslinks. The authors will need to repeat the pull-down experiments after DNase digestion and in the absence of crosslinker in order to rule out these concerns*.

We thank the reviewers for raising this important point. We performed the experiment suggested by the reviewers and found that the amount of the Ycs4 condensin subunit co-purifying with Sgo1 was independent of treatment with cross-linker or DNase. This shows that the interaction between Sgo1 and condensin is not dependent on DNA. This data is now presented in Figure 4—figure supplement 1. We agree that the amount of condensin that purifies with Sgo1 is a small fraction of the total cellular condensin. However, this is expected as our Brn1 ChIP-Seq showed that the bulk of condensin is associated with the rDNA and would not be expected to bind to Sgo1.

*A second argument for a role of Sgo1 in condensin loading is that centromeric condensin levels increase upon Sgo1 overexpression. The increase in condensin binding could, however, result from the delay in anaphase onset caused by Sgo1 overexpression. Would condensin levels remain unchanged if anaphase onset were delayed by other means*?

To exclude the possibility that Brn1’s chromosomal enrichment in *SGO1-*overexpressing cells was a consequence of the metaphase arrest caused by high levels of Sgo1, we overexpressed *SGO1* in cells arrested in mitosis by treatment with nocodazole. Measurement of Brn1 association with two centromeric sites and a pericentromeric site by ChIP-qPCR showed that *SGO1* overexpression also increased Brn1 levels in cells arrested in mitosis, arguing against the idea that the centromeric enrichment of Brn1 is solely due to the cell cycle block caused by *SGO1* overexpression. This data is presented as Figure 6—figure supplement 1 in the revised version.

*Ectopic recruitment of condensin by Sgo1 fused to TetR seems rather inefficient, with a two-fold enrichment (and considerable variation?). The authors should at least test whether ectopic recruitment can be measured at both sides of the TetO repeats. Yet a much better test whether condensin recruitment by Sgo1 is significant would be to employ the TetO-TetR approach to tether to centromeres an Sgo1 mutant that can otherwise not bind (e.g., Sgo1-100 or Sgo1-700) and assay whether this restores condensin levels and chromosome bi-orientation*.

In the original manuscript, we used a primer set ∼800 bp to the left of the *tetO* repeats. Following sonication during the ChIP protocol, the majority of fragments are much shorter than this (in the range of ∼200 bp), so a primer set at this distance is expected to only inefficiently detect proteins recruited to the *tetO* repeats themselves. As suggested by the reviewers, we have now analyzed a site on the other side of the *tetO* repeats which is much closer (∼50bp) to the tethering site. As can be seen in Figure 6 in the revised manuscript, both Brn1 (condensin) and Rts1 (PP2A) are efficiently recruited to this site, and Ipl1 (CPC) less so. This result now provides compelling evidence for the direct recruitment of condensin by Sgo1.

As further suggested by the reviewers, we also tethered the Sgo1-100, Sgo1-700 and Sgo1-3A proteins to the centromere in a strain that otherwise lacks *SGO1*. As we showed in Figure 1, normally, both Sgo1-100 and Sgo1-700 fail to be maintained on centromeres. However, Sgo1-100 can bind condensin, whereas Sgo1-700 cannot (Figure 9). This finding is recapitulated in the tethering experiment. Only Sgo1-100 and not Sgo1-700 is capable of recruiting Brn1 (condensin) to the tethering site. As expected, tethered Sgo1-3A, which localizes to centromeres and associates with condensin is also able to recruit condensin to the tethering site. This data is shown in Figure 9. Together with the finding that only the *sgo1-700* mutant loses the bias to sister kinetochore biorientation (Figure 9), this provides strong evidence that Sgo1 biases sister kinetochores for biorientation through condensin recruitment.

We also tested the ability of the tethered Sgo1 proteins to rescue sister kinetochore biorientation after treating an unsynchronous population with nocodazole and examining segregation of the chromosome to which Sgo1-TetR-GFP is tethered in anaphase cells after nocodazole washout. Similar to the assays in Figures 1 and 7, accurate segregation in this assay would require the activity of the error correction machinery. As expected, due to the impaired ability of all Sgo1 mutant proteins to maintain Ipl1 association with the centromere (Figure 9) only tethered wild type Sgo1 was able to partially rescue the segregation of the chromosome to which it was tethered (chromosome IV; shown in Figure 9—figure supplement 1). For technical reasons (both *SGO1-tetR-GFP* and *tetR-tdTomato* are integrated into the same genomic locus), it was necessary to perform this experiment in diploid cells and therefore we cannot presently test the ability of these tethered constructs to retain sister kinetochore bias (we cannot easily arrest and release the cells from G1 as they are not sensitive to mating pheromone). Moreover, other unpublished findings in the lab have suggested that the dynamicity of Sgo1 on centromeres is key to its regulation and important for accurate segregation so it is likely that Sgo1 tethering affects its function. Indeed, even tethered wild type Sgo1-tetR-GFP is not fully able to rescue the mis-segregation of chromosome IV in our experiment (Figure 9—figure supplement 1). Nevertheless, our demonstration of a lack of sister kinetochore bias in the *sgo1-700* mutant, which cannot bind condensin (Figure 9) or recruit it to centromeres (Figure 9) provides compelling evidence that the Sgo1-condensin interact is critical in biasing sister kinetochores to biorient.

*An explanation for the altered binding profiles of condensin measured by ChIP-seq might be changes in cohesin accumulation at centromeres in the Sgo1 mutants. The authors need to check by ChIP-qPCR (and ideally also ChIP-seq) whether cohesin levels at centromeres are affected in the Sgo1 mutant*.

We performed both these experiments and the data is shown in Figure 5 and associated figure supplements. Cohesin association with chromosomes is indistinguishable in wild type and *sgo1*? cells.

Overall, we feel our study now argues strongly for a direct role of Sgo1 in condensin recruitment to the pericentromere.

*2) The authors monitor the splitting of a sister centromere pair as a measure for sister chromatid bi-orientation (*Figures 1 and 5*,*
Figure 3—figure supplement 2*). The reduction of sister centromere splitting observed in the Sgo1 and condensin mutants could, however, equally well result from an increase in centromeric sister chromatid cohesion. An increase in centromeric cohesion would prevent splitting of bi-oriented sister chromatids*.

*One possibility to distinguish whether chromatid pairs have attached to spindle microtubules from only one spindle pole or whether chromatids have bi-oriented but their centromeres cannot separate would be to score centromere-spindle pole body (SPB) distances in the live cell microscopy assay (*Figure 5*). If centromeres were bi-oriented but merely failed to split, they should be positioned at equal distances to both SPBs. If sister centromeres failed to bi-orient, they should frequently be found close to one SPB. Alternatively, the authors could release cells from the Cdc20 arrest and allow them to enter anaphase, then score whether sister centromeres segregate to one pole or to opposite poles (see Indjeian and Murray, CurBiol 2007)*.

As mentioned above and below, we have obtained no evidence that Sgo1 affects cohesion in budding yeast mitosis. Importantly, the distance between separated sister centromeres in *sgo1*? and *YCS5-aid* cells is not significantly different to that of wild type cells (Figure 5). This is strong evidence that cohesion is not affected in these cells and that the decreased frequency of sister centromere is due to defective sister-kinetochore biorientation. Furthermore, as suggested by the reviewers, we measured the distance from *CEN4-GFP* foci to the nearest SPB, in *sgo1*?, *YCS5-aid* and *ipl1-as* cells with a single *CEN4-GFP* focus after nocodazole washout (shown in Figure 9—figure supplement 1). Taking the median *CEN4-SPB* distance for wild type as the typical distance, we scored the percentage of cells with shorter or longer distances in each case. In all three mutants, the fraction of cells with a *CEN4-SPB* shorter than the wild type median was increased, indicating that unpaired sister centromeres tend to reside closer to the SPB, supporting the idea that chromosomes fail to align in the centre of the spindle, as expected when biorientation fails. This effect was less pronounced in the condensin mutant, but this is expected as this has a milder biorientation defect overall (Figure 7—figure supplement 1). Indeed, Ipl1 is localized in *YCS5-aid* cells (Figure 7) and the error correction process is partially functional (Figure 7).

The mis-segregation of chromosomes following release from nocodazole treatment into anaphase is well established to occur in *sgo1* mutants (11; 22) and we have recapitulated these findings in Figure 9—figure supplement 1, using tethered *sgo1* alleles. However, similar experiments with condensin mutants are difficult to interpret as condensin is critical for chromosome arm segregation in anaphase (Renshaw et al., Dev Cell 2010). The situation is further complicated by the fact that due to the partial functionality of the error correction machinery in condensin mutants (Figure 7), condensin is unlikely to be essential for sister kinetochore biorientation, but rather provide a bias that facilitates sister kinetochore capture and efficient error correction. Analysis of the *sgo1-700* mutant that fails to associate with condensin (Figure 7) provides strong evidence that Sgo1 promotes the bias to sister kinetochore biorientation through condensin recruitment.

*The data in*
Figures 6 and 7
*show a mild, real defect in biorientation but don't specifically address the of kineteochore geometry. The authors should either perform further experiments to support their hypothesis or confine their “geometric bias” to speculation in the Discussion*.

We agree with the reviewers that condensin mutants show a biorientation defect. We also agree that direct evidence for a role of condensin in kinetochore geometry is lacking. We have therefore restricted our references to a possible role in geometry to speculation.

*3) Most of the graphs that represent the results from ChIP-qPCR experiments (*Figures 1, 2, 5 and 7*,*
Figure 5—figure supplement 3*) lack information that is essential to evaluate the data. What do the column plots represent? Do they show mean values, and if yes, from how many PCR runs and how many independent immunoprecipitation experiments? What do the error bars represent? The legend for*
Figure 4
*is missing completely*.

We apologize for this oversight. We have now included a more detailed description of the procedures used in the Materials and methods and detailed figure legends throughout, explaining how each experiment was carried out, the procedures used to process the data and the meaning of the figures presented. As detailed in the figure legends, the vast majority of ChIP-qPCR experiments show mean values of three independent repeats. qPCR was always performed in triplicate (this information is now included in the Materials and methods). The number of independent immunoprecipitation experiments is given in the figure legends (typically 3). Error bars represent standard error. The exception to this are time course experiments where the day-to-day variability of cell cycle progression means data from different days cannot be combined usefully. Here, experiments are repeated, typically 3 times in total, and a representative experiment is shown. Overall, the pattern does not vary, though the absolute values at each timepoint can.

*Taking into account the possible variability between individual ChIP-qPCR experiments, careful statistical analysis will be required to evaluate the significance of the observed differences. For example, the values measured at CEN4 for the same wild-type strain (AM3513) in two independent Ipl1 HA-ChIP experiments vary 4-fold (compare*
Figure 2*), calling into question whether the 2-fold reduction in ChIP signals in the Sgo1 mutants (*Figure 2*) is significant. Similarly, the ChIP signal at CEN4 measured for the Sgo1-100 mutant in G1 phase is more than 2-fold higher than the signal for wildtype Sgo1 (*Figure 1*). Is the 4-fold increase in Sgo1 ChIP signals during mitosis (e.g., 2-fold when compared to the Sgo1-100 mutant) then still significant*?

Again, we apologize: these discrepancies are due to our failure to provide adequate figure legends describing the procedures used to obtain the data and can all be accounted for by different growth conditions. In particular, the ChIP-qPCR signals obtained in experiments where nocodazole was directly added to cycling cells are typically much higher than experiments where cells were first arrested in G1 before being released into nocodazole. The fraction of cells that were approaching, or in, mitosis at the time of nocodazole will spend a much greater time in mitosis, where our proteins of interest are presumably continually recruited, thereby accounting for these differences. Of course, repeats are carried out using exactly the same conditions, with all strains within one repeat grown and processed on the same day, allowing direct comparison for those conditions. Specifically, for the experiment shown in Figure 2 in the old version (now Figure 3), cells were first arrested in G1 before release into nocodazole (this was to allow Sgo1-aid degradation in a single cell cycle), whereas cells for the experiment shown in Figure 2 in the old version (now 3E) were arrested directly in nocodazole. The difference between the levels of Sgo1 and *Sgo1-100* mutant proteins is likely due to slight changes in the timing of alpha factor release in the handling of these cultures (see above about the interpretation of time course experiments). The main point of these experiments was to show that residual association to centromeres occurs even though it is not maintained in nocodazole-arrested cells. We do not think it is appropriate to take any further interpretation based on the absolute values from these experiments as due to the inherent variability of time course experiments; to do so would require exquisitely controlled conditions and much more closely spaced sampling.